

# Hydrodynamics in lattice models
# with continuous non-Abelian symmetries

**Paolo Glorioso**[1*], **Luca V. Delacrétaz**[1†], **Xiao Chen**[2],
**Rahul M. Nandkishore**[3] and **Andrew Lucas**[3‡]

**1** Kadanoff Center for Theoretical Physics, University of Chicago, Chicago, IL 60637
**2** Department of Physics, Boston College, Chestnut Hill, MA 02467, USA
**3** Department of Physics and Center for Theory of Quantum Matter,
University of Colorado, Boulder CO 80309, USA

★ paolog@stanford.edu, † lvd@uchicago.edu, ‡ andrew.j.lucas@colorado.edu

## Abstract

We develop a systematic effective field theory of hydrodynamics for many-body systems on the lattice with global continuous non-Abelian symmetries. Models with continuous non-Abelian symmetries are ubiquitous in physics, arising in diverse settings ranging from hot nuclear matter to cold atomic gases and quantum spin chains. In every dimension and for every flavor symmetry group, the low energy theory is a set of coupled noisy diffusion equations. Independence of the physics on the choice of canonical or microcanonical ensemble is manifest in our hydrodynamic expansion, even though the ensemble choice causes an apparent shift in quasinormal mode spectra. We use our formalism to explain why flavor symmetry is qualitatively different from hydrodynamics with other non-Abelian conservation laws, including angular momentum and charge multipoles.

As a significant application of our framework, we study spin and energy diffusion in classical one-dimensional SU(2)-invariant spin chains, including the Heisenberg model along with multiple generalizations. We argue based on both numerical simulations and our effective field theory framework that non-integrable spin chains on a lattice exhibit conventional spin diffusion, in contrast to some recent predictions that diffusion constants grow logarithmically at late times. We show that the apparent enhancement of diffusion is due to slow equilibration caused by (non-Abelian) hydrodynamic fluctuations.



# 1 Introduction

Hydrodynamics is a universal language for describing thermalization and slow dynamics in chaotic many-body systems, whether they are classical or quantum [1].[1] The universality of hydrodynamics arises from its insensitivity to nearly all microscopic details, except for space-time symmetries and conservation laws.

This paper is about the hydrodynamics of systems with continuous non-Abelian symmetries. Physical realizations include flavor charge in quark-gluon plasma [4] (or color charge at high temperatures), spin-orbit coupled solid-state systems [5, 6], and SU(2)-symmetric cold atomic gases [7, 8]. The "non-Abelian hydrodynamics" of these systems has been studied by many authors, for many decades [9–15]. However, in our view, the literature can be unclear

---

[1]Note that there is also the subject of generalized hydrodynamics [2, 3], relevant to integrable systems; this is not the subject of the present paper.

and can even appear contradictory about elementary questions, including the number of hydrodynamic degrees of freedom!

In light of recent developments in effective field theory which allow, at long last, for a systematic derivation of the effective action of hydrodynamics from a Schwinger-Keldysh (quantum) field theory, we present here a systematic derivation of hydrodynamics in models with non-Abelian flavor symmetry. We specifically focused on lattice models where momentum is not conserved, which allows us to focus on the new effects arising from the non-Abelian flavor symmetry. Our derivation emphasizes a number of points, which have appeared in previous literature, but are here derived systematically and from a clear set of postulates. (*1*) A flavor symmetry group with $r$ generators contributes $r$ independent hydrodynamic modes, at all orders in the hydrodynamic derivative expansion (Section 2.2). (*2*) Hydrodynamics in microcanonical and canonical ensembles describe the same physics, even though the canonical ensemble appears to modify the location of measurable poles in Green's functions (Section 2.3). (*3*) The hydrodynamic modes of any lattice model whose conserved charges are energy and non-Abelian flavor charges are of the form $\omega = \pm Ak^2 - iDk^2 + \cdots$, with $D > 0$ and $A$ nonzero only at finite flavor charge density (Section 3). This conclusion is robust in nonlinear fluctuating hydrodynamics, in all spatial dimensions $d$, including $d = 1$. (*4*) The non-Abelian nature of flavor symmetry has qualitatively different implications than other non-Abelian symmetries in hydrodynamics, such as rotational invariance or multipole symmetries (Section 4). Some of these confusions arose because of unclear treatments of global vs. gauge symmetries, and because turning on background gauge fields $A$ is more drastic with non-Abelian degrees of freedom (equations of motion depend on $A$, not just its derivatives).

Our main motivation for revisiting the derivation above is recent literature on classical and quantum spin chains with SU(2) symmetry in $d = 1$ [16–35] (Section 6). In fact, we were inspired to carry out this effective field theory computation in part by papers arguing that the infinite temperature classical Heisenberg model does not have conventional spin diffusion [16, 24, 32], as predicted and obtained by many other authors [18, 21, 30]. In particular, [32] argues that the spin diffusion constant scales as

$$D(t) \sim \log^{4/3} t. \tag{1}$$

Remarkably, we have found certain models of nonlinear fluctuating hydrodynamics, with sufficiently many conserved charges, which reproduce logarithmically-enhanced diffusion constants (Section 5). However, the mechanism is not related to non-Abelian flavor symmetry groups. We have also performed extensive numerical simulations which suggest that the only effective theory compatible with the chaotic Heisenberg model is vanilla diffusion with a finite diffusion constant at late times. On the other hand, our numerics does show an apparent enhancement of diffusion compatible with (1): we show in Sec. 6.4 that the effect is due to hydrodynamic fluctuations captured by the effective field theory of Sec. 3. We expect that our method of analysis will prove useful in analyzing dissipative physics in other one dimensional spin models.

This paper is, for the most part, written in a modular way. In particular, our numerical tests on hydrodynamics in spin chains of Section 6 can qualitatively be understood independently of the other Sections.

# 2 Preliminaries

## 2.1 Non-Abelian continuous symmetry

In this paper, we are interested in the many-body dynamics of chaotic lattice models with a non-Abelian flavor symmetry. We will define these terms precisely for a quantum system;

analogous definitions exist for classical systems.

Let $\mathcal{H}$ be the Hilbert space of a many-body system, which we assume can be built up out of the individual quantum degrees of freedom on every site of a lattice $\Lambda$:

$$\mathcal{H} = \bigotimes_{x \in \Lambda} \mathcal{H}_x. \tag{2}$$

Quantum dynamics corresponds to a one parameter unitary transformation $U(t)$ on $\mathcal{H}$: for example, $U(t) = \mathrm{e}^{-iHt}$ if there exists a time-independent Hamiltonian $H$. We do not assume in this paper that such a Hamiltonian exists.

Let $G$ be a (compact) Lie group. We study quantum dynamical systems $U(t)$ with symmetry $G$ subject to: (1) there exists a representation of $G$ – a set of unitary matrices $V(g)$ for each group element $g \in G$ – acting on $\mathcal{H}$, such that

$$[U(t), V(g)] = 0 \ \text{ for all } g \in G \text{ and } t \in \mathbb{R}, \tag{3}$$

and (2) the unitary $V(g)$ may further be expanded as

$$V(g) = \bigotimes_{x \in \Lambda} V_x(g), \tag{4}$$

where $V_x(g)$ are unitary matrices acting on the individual Hilbert spaces.

It is conventional to focus on group elements $g$ infinitesimally close to the identity. Schematically writing $g = 1 + \epsilon^A T^A$, where $A = 1, \ldots, \dim(G)$, we may define the charge per site $q_x^A$ by

$$V_x(1 + \epsilon^A T^A) = 1 + i\epsilon^A q_x^A, \tag{5}$$

along with total charge

$$Q^A = \sum_{x \in \Lambda} q_x^A. \tag{6}$$

Combining the above equations implies that

$$[U(t), Q^A] = 0. \tag{7}$$

Therefore, if we have a quantum system in initial mixed state $\rho(0)$, which time evolves to $\rho(t) = U(t)\rho(0)U^\dagger(t)$, then

$$\frac{\mathrm{d}}{\mathrm{d}t}\mathrm{tr}\big(\rho(t)Q^A\big) = \frac{\mathrm{d}}{\mathrm{d}t}\big\langle Q^A(t)\big\rangle = 0. \tag{8}$$

All charges are conserved.

What is non-trivial about non-Abelian groups is that there exist a set of fully antisymmetric structure constants $f^{ABC}$ (at least one of which is non-vanishing), such that

$$[Q^A, Q^B] = if^{ABC}Q^C. \tag{9}$$

We emphasize that there is no contradiction between (8) and (9). Yet, in some sense, there is an intuitive tension about how (8) and (9) might both arise in hydrodynamics, where we aim to describe the slow dynamics of the densities of conserved quantities. How can we keep track of all the densities of conserved charges, if no quantum state actually has a definite charge (eigenvalue) under all $Q^A$?

We will be studying systems with parity (P) and time-reversal (T) symmetry. Because these are respectively unitary and antiunitary, they are embedded in the algebra (9) as (note that $f^{ABC} \in \mathbb{R}$)

$$\mathsf{P}^{-1}Q^A\mathsf{P} = Q^A, \qquad\qquad \mathsf{T}^{-1}Q^A\mathsf{T} = -Q^A, \tag{10}$$

i.e. the charges are parity even and time-reversal odd. For some specific groups, there are other ways to embed these discrete symmetries in the algebra: for example if $G = G_\mathrm{L} \times G_\mathrm{R}$, then the left and right charges can map into each other, e.g. $\mathsf{P}^{-1}Q_\mathrm{L}^A\mathsf{P} = Q_\mathrm{R}^A$. However (10) is the only possibility for simple Lie groups which we will be focusing on.

## 2.2 Hydrodynamic decomposition of Hilbert space

Before developing a classical effective theory – hydrodynamics – for a thermalizing system with a non-Abelian flavor symmetry, we must carefully resolve the question of how many hydrodynamic degrees of freedom there are. We will see that all charges $Q^A$ represent *independent* hydrodynamic degrees of freedom, and that, in a domain of length $L$, this conclusion will be robust to all orders in the perturbative hydrodynamic expansion in $L^{-1}$. This conclusion can be found in [12, 13], but appears to conflict with other literature [5].

We note that an asymptotic hydrodynamic expansion to arbitrary derivative orders does not mean that there is not a very long (but finite) time scale before which hydrodynamics does not make sense. This is the regime in which "quasihydrodynamic" models are more appropriate (following the terminology of [36]); it has been observed in, e.g., [37], that such quasihydrodynamic regimes can exist on intermediate time scales in spin chains. There is a more subtle question of long-time tails, whereby hydrodynamic fluctuations themselves can break the gradient expansion; we will see that this indeed happens in the models studied in this paper. Still, the presence of long time tails does not invalidate the hydrodynamic framewor.

For pedagogy, we focus the discussion on lattice models with SU(2) symmetry. We assume that each lattice site has a two-level, spin-$\frac{1}{2}$, degree of freedom. In a box of $L$ lattice sites, there are $2^L$ states in Hilbert space. Within this box, we assume the three conserved non-Abelian charges $Q^A$, and their corresponding densities $n^A$, are simply the total spin:

$$Q^A = L n^A = \sum_{i=1}^{L} \sigma_i^A, \tag{11}$$

where $[\sigma_i^A, \sigma_j^B] = \delta_{ij} \mathrm{i} \epsilon^{ABC} \sigma_j^C$ are one-site Pauli matrices. Now, observe that

$$[n^A, n^B] = \frac{\mathrm{i}}{L} \epsilon^{ABC} n^C. \tag{12}$$

In the hydrodynamic limit $L \to \infty$, all densities approximately commute.

This Hilbert space of $L$ spins can then be approximately decomposed (when $n^A n^A < 1$) into subsectors corresponding to each density $n^A$. Denoting $\mathbb{P}(n^A)$ as a projector onto the Hilbert space of spin density $n^A$ (within this subsector, we can approximate the operators $n^A$ as constant) we find that

$$\frac{\mathrm{tr}(\mathbb{P}(n_1^A)\mathbb{P}(n_2^A))}{\sqrt{\mathrm{tr}(\mathbb{P}(n_1^A))\mathrm{tr}(\mathbb{P}(n_2^A))}} \sim \exp\left[-\frac{9}{8} L \left|n_1^A - n_2^A\right|^2\right], \tag{13}$$

when $|n_1^A|, |n_2^A| \ll 1$. The technical construction is provided in Appendix A. On long time scales, so long as we are only interested in correlation functions of the density operators $n^A$, we may approximate the many-body density matrix within one box by

$$\rho(t) \approx \int \mathrm{d} n^A \, p(n^A, t) \frac{\mathbb{P}(n_1^A)}{\mathrm{tr}(\mathbb{P}(n_1^A))}, \tag{14}$$

where $p(n^A, t)$ can be interpreted as the classical probability density of the charge density $n^A$. As we can do this process in every box $x$ of size $L$, the many-body Schrödinger equation will then become a Fokker-Planck equation for a stochastic $n^A(x, t)$. Following the logic of [38], we take the resulting stochastic equations as the definition of nonlinear fluctuating hydrodynamics. While there is not a universal proof that such equations are equivalent to those more commmonly derived by effective field theory methods (as below), we do not know of a counter-example in a thermalizing and chaotic system!

Our technical detour teaches us that in the hydrodynamic limit, we should indeed treat every non-commuting charge density as a separate hydrodynamic degree of freedom. $p(n^A, t)$ is well-defined up to fluctuations in $n^A$ of order $L^{-1/2}$. In a conventional hydrodynamic theory with commuting conserved charges, fluctuations of this order are also present [38], and arise from finite-size statistical fluctuations in the conserved quantities. We thus do not foresee the gradient expansion being any worse behaved for the theory with non-commuting charges than in a conventional hydrodynamic theory.

## 2.3 Microcanonical vs. canonical ensembles

Having confirmed our hydrodynamic degrees of freedom correspond to densities $n^A$ associated with each conserved charge $Q^A$ from flavor symmetry $G$, we now turn to one more subtle technical issue. In ordinary thermodynamics and hydrodynamics, in chaotic many-body systems, physics does not depend on whether we study a microcanonical or (grand) canonical ensemble. Let us consider the grand canonical ensemble generated by sourcing the non-Abelian charges $Q^A$ with chemical potentials $\mu^A$. In a many-body language, this corresponds to deforming a static Hamiltonian by

$$H \to H - \mu_{\text{ext}}^A Q^A; \tag{15}$$

in the effective theory language, this corresponds to turning on a background, non-dynamical gauge field

$$A^A = \mu_{\text{ext}}^A dt. \tag{16}$$

In each case, we have used the 'ext' subscript to emphasize that this is not a "local chemical potential" as seen by the fluid – it is the coefficient of an external source coupling to the theory. Using manipulations analogous to Appendix A, it is straightforward to show that $\mu^A$ indeed drives the system to an average density $n^A$. The precise nonlinear relation between $\mu$ and $n$ forms the thermodynamic equation of state.

When the symmetry is non-Abelian, the seemingly benign switch to a canonical ensemble (rather than restricting to an initial density matrix of fixed average density $n^A$) causes an unwanted effect: it explicitly violates conservation laws! Now we find that

$$\frac{d}{dt}\langle Q^A\rangle = \langle i[H(t) - \mu_{\text{ext}}^B Q^B, Q^A]\rangle = -\mu_{\text{ext}}^B f^{ABC}\langle Q^C\rangle. \tag{17}$$

In the local hydrodynamic language, which we will detail in Section 3, the local density obeys

$$\partial_t n^A + \partial_i J^{iA} = -\mu_{\text{ext}}^B f^{ABC} n^C, \tag{18}$$

where $J^{iA}$ denotes the spatial components of the flavor charge current.

However, this global violation of conservation laws can be undone. Consider the following change of variables: defining the matrix

$$\mathcal{U}^{AC} = f^{ABC}\mu_{\text{ext}}^B \tag{19}$$

we define $\tilde{n}^A$ by

$$\tilde{n}^A = \exp[\mathcal{U}t]^{AC} n^C. \tag{20}$$

Plugging this in to (18), we see that

$$\partial_t \tilde{n}^A + \partial_i J^{iA} = 0. \tag{21}$$

Hence, so long as $J^{iA}(\tilde{n}^B) = J^{iA}(n^B)$, the hydrodynamic equations are not changed by this time-dependent field redefinition. Indeed, we will see in Section 3 that the consistency of hydrodynamics ensures such invariance, confirming the equivalence between canonical and

microcanonical ensembles. In atomic physics, (20) is referred to as transforming into the "rotating frame".

We emphasize that a necessary feature of hydrodynamics, in any theory with non-Abelian conserved charges, is that the hydrodynamic modes in the canonical ensemble are *exactly untouched*, up to a global "rotation" set by the external chemical potential. As an example, we can consider a theory with SU(2) symmetry, with a finite charge density in the $z$-direction. We will see in Section 3 that the $x, y$ components of charge have dispersion relation

$$\omega_{\text{microcanonical}}(k) = \pm A k^2 - i D_\perp k^2 + \cdots. \tag{22}$$

In the canonical ensemble, to all orders in hydrodynamics, we must find

$$\omega_{\text{canonical}} = \pm(\mu_{\text{ext}} + A k^2) - i D_\perp k^2 + \cdots. \tag{23}$$

While normally, the presence of a $k$-independent term would imply that this mode is simply non-hydrodynamic, a key feature of flavor hydrodynamics is that $\omega_{\text{canonical}}$ is a hydrodynamic mode, precisely because the transformation (20) allows us to set $\mu_{\text{ext}} = 0$. Assuming that $\omega_{\text{microcanonical}}$ is evaluated at the same average density as $\omega_{\text{canonical}}$ will be no further $\mu_{\text{ext}}$ corrections within (23). Our systematic field theory will indeed produce the necessary result (23).

# 3 Effective field theory of hydrodynamics

## 3.1 Overview of the method

Hydrodynamics is generally presented as a set of equations describing the long time dynamics of conserved charges. In this Section we present a systematic effective field theory of hydrodynamics, where the latter is introduced in terms of an action principle uniquely specified in terms of basic symmetry principles. This approach has several advantages. First, the traditional formulation of hydrodynamics is subject to various constraints that are imposed at phenomenological level, such as the second law of thermodynamics and Onsager relations [39]. In the effective field theory adopted below, these constraints arise naturally as a consequence of a particular symmetry of the effective action, along with basic properties which are a remnant of the conservation of probability (i.e., unitarity).[2] The second advantage is that this effective field theory systematically captures any term contributing to the low-energy dynamics, including non-Gaussianities of the noise. In more traditional approaches to stochastic hydrodynamics such as the Martin-Siggia-Rose (MSR) formalism [41], the noise is introduced by demanding consistency with the fluctuation-dissipation theorem. Such procedure works well when the equations are linear in the noise. In our approach, nonlinear dynamics, including that of the noise, is manifestly compatible with all symmetries, including Kubo-Martin-Schwinger invariance. While many of our main results in Section 6 turn out to agree with the predictions of the MSR approach, given the various debates in previous literature that we aim to address here, we have opted for the most careful derivation of nonlinear fluctuating hydrodynamics that we know.[3] We will use the systematic nature of this effective field theory in later Sections to rule out any anomalous hydrodynamic transport behavior of spin diffusion for non-integrable SU(2)-invariant spin chains. Below we shall give an overview of the formalism developed in [43, 44] (see also [45, 46], and [47] for a review) using the simplest example of an ordinary conserved U(1) current. We will extend it to the non-Abelian case in the next Section.

---

[2]See [40] for a thorough comparison between the constraints coming from the second law and those coming from the effective action approach.

[3]An example of terms that cannot be captured in the MSR formalism was recently studied in [42].

Consider a system with dynamics described by a Hamiltonian $H$ which is invariant under a global U(1) symmetry. We denote the associated conserved current by $J^\mu = (J^t, J^i)$. We take the initial state to be in the local Gibbs ensemble $\rho_0 = e^{-\beta(H-\mu_0 Q)}/\text{tr}(e^{-\beta(H-\mu_0 Q)})$, where $Q = \int_x J^t$ is the total charge and $\mu_0 = \mu_0(\vec{x})$ is the initial chemical potential, taken to be a slowly-varying function of $\vec{x}$. The Schwinger-Keldysh generating functional for the current correlation functions is

$$e^{iW[A_{1\mu}, A_{2\mu}]} = \text{tr}\left[\mathcal{T}\left(e^{-i\int_{t_i}^{t_f}(H-\int_x A_{1\mu}J^\mu)}\right)\rho_0\bar{\mathcal{T}}\left(e^{i\int_{t_i}^{t_f}(H-\int_x A_{2\mu}J^\mu)}\right)\right]$$
$$= \int_{\rho_0} D\psi_1 D\psi_2\, e^{iS_0[\psi_1, A_{1\mu}]-iS_0[\psi_2, A_{2\mu}]}, \tag{24}$$

where $\mathcal{T}, \bar{\mathcal{T}}$ denote time- and anti-time ordering, and we coupled the Hamiltonian in the forward and backward evolutions to background gauge fields $A_{1\mu}$ and $A_{2\mu}$, respectively. Varying with respect to $A_{1\mu}$ ($A_{2\mu}$) brings down time-ordered (anti-time ordered) insertions of the current $J^\mu$. Current conservation implies the Ward identity

$$W[A_{1\mu} + \partial_\mu\lambda_1, A_{2\mu} + \partial_\mu\lambda_2] = W[A_{1\mu}, A_{2\mu}], \tag{25}$$

where $\lambda_1(t, \vec{x})$ and $\lambda_2(t, \vec{x})$ are two independent functions. In the last expression in (24) we wrote the forward and backward evolutions in terms of a path integral over a doubled copy of the degrees of freedom collectively denoted by $\psi_1$ and $\psi_2$, where $S_0$ denotes the microscopic action of the system. The boundary conditions for $\psi_1, \psi_2$ at the initial time $t_i \to -\infty$ account for the presence of the initial state. The boundary conditions at the final time $t_f \to \infty$ consist in identifying the doubled fields $\psi_1(t_f) = \psi_2(t_f)$ and are due to the fact that we are taking the trace. These boundary conditions are the sole coupling between the two copies of the fields.

Due to the presence of long-living modes associated to the conservation of $J^\mu$, the generating functional $W[A_{1\mu}, A_{2\mu}]$ will be nonlocal. The proposal of [43] is to "integrate in" such modes responsible for this nonlocality. Let us denote these long-living modes by the fields $\varphi_1(t, \vec{x})$ and $\varphi_2(t, \vec{x})$, whose nature will become clear momentarily. The nonlocal generating functional $W$ can then be written as the path-integral of a *local* effective action:[4]

$$e^{iW[A_{1\mu}, A_{2\mu}]} = \int D\varphi_1 D\varphi_2\, e^{iS_{\text{eff}}[A_{1\mu}, \varphi_1, A_{2\mu}, \varphi_2]}. \tag{26}$$

The statement that $\varphi_1, \varphi_2$ are associated to the conservation of $J_1^\mu, J_2^\mu$ translates into the fact that, upon integrating out such degrees of freedom, the resulting $W$ should satisfy the Ward identity (25). Equivalently, each of the two currents, obtained by varying $S_{\text{eff}}$ with respect to the corresponding sources,[5]

$$J_1^\mu \equiv \frac{\delta S_{\text{eff}}}{\delta A_{1\mu}}, \qquad J_2^\mu \equiv -\frac{\delta S_{\text{eff}}}{\delta A_{2\mu}}, \tag{27}$$

should be conserved upon solving the equations of motion of $\varphi_1, \varphi_2$ when performing the path integral (26) in the saddle-point limit. This implies that the effective action $S_{\text{eff}}$ should depend on various fields through the following combinations:[6]

$$B_{1\mu} = \partial_\mu\varphi_1 + A_{1\mu}, \qquad B_{2\mu} = \partial_\mu\varphi_2 + A_{2\mu}, \tag{28}$$

---

[4]In some cases, additional long-living modes might be present, e.g. slowly moving order parameter. These should be included in the effective action as independent degrees of freedom. For simplicity, we shall not consider such situation here.

[5]The relative minus sign comes from (24).

[6]In the presence of anomalies, the dependence of the action on $A_\mu$ and $\partial_\mu\varphi$ is slightly modified [48].

i.e. $S_{\text{eff}} = S_{\text{eff}}[B_{1\mu}, B_{2\mu}]$. The combinations (28) lead to naturally interpret $\varphi_1, \varphi_2$ as the parameters of the gauge transformations in (25). However, at the level of $S_{\text{eff}}$, $\varphi_1, \varphi_2$ are dynamical fields as current conservation does not hold before extremization of the effective action. Unlike in (24), the action in (26) does not have a factorized structure: the two copies of slow variables now interact locally as a consequence of having integrated out the fast degrees of freedom [49]. These cross-couplings characterize dissipation and fluctuations. As a consequence of unitarity of the underlying system, the effective action must satisfy the following properties:

$$S_{\text{eff}}[\varphi, \varphi; A_\mu, A_\mu] = 0, \qquad S_{\text{eff}}[\varphi_2, \varphi_1; A_{2\mu}, A_{1\mu}] = -S_{\text{eff}}^*[\varphi_1, \varphi_2; A_{1\mu}, A_{2\mu}],$$
$$\text{Im}\, S[\varphi_1, \varphi_2; A_{1\mu}, A_{2\mu}] \geq 0 \,, \tag{29}$$

which can be proven by comparing (26) with (24), and using unitarity of the evolution operator $\mathcal{T}(e^{-i\int_{t_i}^{t_f}(H - \int_x A_\mu J^\mu)})$ [50]. Furthermore, assuming that the Hamiltonian $H$ is invariant under the composition of parity and time reversal PT, and since the system is in local thermal equilibrium, the effective action satisfies dynamical KMS invariance, i.e.:

$$S_{\text{eff}}[\varphi_1, \varphi_2; A_{1\mu}, A_{2\mu}] = S_{\text{eff}}[\tilde{\varphi}_1, \tilde{\varphi}_2; \tilde{A}_{1\mu}, \tilde{A}_{2\mu}] \,, \tag{30}$$

with

$$\tilde{\varphi}_1(t, \vec{x}) = (-1)^\eta \varphi_1(-t, P\vec{x}), \qquad \tilde{\varphi}_2(t, \vec{x}) = (-1)^\eta \varphi_2(-t - i\beta, P\vec{x}) \tag{31a}$$
$$\tilde{A}_{1\mu}(t, \vec{x}) = (-1)^{\eta_\mu} A_{1\mu}(-t, P\vec{x}), \qquad \tilde{A}_{2\mu}(t, \vec{x}) = (-1)^{\eta_\mu} A_{2\mu}(-t - i\beta, P\vec{x}) \,, \tag{31b}$$

where $(-1)^\eta = \pm 1$ and $(-1)^{\eta_\mu} = \pm 1$ are the PT eigenvalues of $\varphi$ and $A_\mu$, respectively, and where P acts on an odd number of spatial coordinates, e.g. $P\vec{x} = (-x_1, x_2, \ldots, x_d)$. Here we chose PT for concreteness; Eqs. (30),(31a),(31b) can be generalized to when the Hamiltonian $H$ is invariant under T, or any other discrete transformation that contains T. This symmetry encodes the periodicity of $n$-point functions along the Euclidean time direction. A derivation of (30) is outlined in Appendix C. Using (28), eqs. (31a),(31b) can be repackaged in terms of $B_\mu$:

$$S_{\text{eff}}[B_{1\mu}, B_{2\mu}] = S_{\text{eff}}[\tilde{B}_{1\mu}, \tilde{B}_{2\mu}] \tag{32}$$
$$\tilde{B}_{1\mu}(t, \vec{x}) = (-1)^{\eta_\mu} B_{1\mu}(-t, P\vec{x}), \qquad \tilde{B}_{2\mu}(t, \vec{x}) = (-1)^{\eta_\mu} B_{2\mu}(-t - i\beta, P\vec{x}) \,. \tag{33}$$

At this point, the effective action can depend on any combination of $B_{1\mu}$ and $B_{2\mu}$ that satisfies (29) and (32). Interestingly, this type of action describes a superfluid, essentially because it depends on both $\partial_t \varphi$ and $\partial_i \varphi$. We are however interested in a phase where the U(1) global symmetry is not spontaneously broken. To ensure this, we require the effective action to be invariant under an additional symmetry, which acts diagonally on the two "phase fields":

$$\varphi_1 \to \varphi_1 + \lambda(\vec{x}), \qquad \varphi_2 \to \varphi_2 + \lambda(\vec{x}) \,, \tag{34}$$

where $\lambda(\vec{x})$ is an arbitrary function of space, and where the background fields $A_{1\mu}, A_{2\mu}$ do not transform. This symmetry is the statement that the value of the diagonal part of the phase at a given time is not physical, and thus spontaneous symmetry breaking cannot occur. Eqs. (29), (32) and (34) constitute the full list of conditions that the effective action $S_{\text{eff}}$ should possess in order to describe the fluctuating hydrodynamics of the conservation of $J^\mu$. We will see concrete expressions of $S_{\text{eff}}$ in the next Section.

Finally, as we are interested in the classical regime of fluctuating hydrodynamics, we can neglect the contribution of quantum effects in the effective action which will in turn simplify part of the calculations. To take the classical limit we restore factors of $\hbar$ and write

$$\varphi_1 = \varphi_r + \frac{\hbar}{2}\varphi_a, \qquad \varphi_2 = \varphi_r - \frac{\hbar}{2}\varphi_a,$$

$$A_{1\mu} = A_{r\mu} + \frac{\hbar}{2}A_{a\mu} = A_{r\mu} - \frac{\hbar}{2}A_{a\mu}, \qquad B_{1\mu} = B_{r\mu} + \frac{\hbar}{2}B_{a\mu} = B_{r\mu} - \frac{\hbar}{2}B_{a\mu} . \tag{35}$$

The $r$- and $a$-fields are often referred to as "classical" and "noise" variables, respectively. This comes from that the $a$-fields do not contribute to the dynamics at mean-field level: they are responsible for the noise. In particular, using (27) we write

$$J_r^\mu \equiv \frac{1}{2}(J_1^\mu + J_2^\mu) = \frac{\delta S_{\text{eff}}}{\delta A_{a\mu}} = J_{(\text{mean-field})}^\mu + \cdots , \tag{36}$$

where the dots stand for terms that contain at least one power of $B_{a\mu}$, and $J_{(\text{mean-field})}^\mu$ depends only on $B_{r\mu}$. At mean-field level, the hydrodynamic equation for current conservation is simply $\partial_\mu J_{(\text{mean-field})}^\mu = 0$. In writing effective actions below we will always use the combinations $B_{r\mu}$ and $B_{a\mu}$ (instead of $B_{1\mu}$ and $B_{2\mu}$) as they are very convenient in the classical limit. Finally, substituing $\beta \to \hbar\beta$ in (32), taking $\hbar \to 0$ we obtain a local transformation

$$\tilde{B}_{r\mu}(t,\vec{x}) = (-1)^{\eta_\mu}B_{r\mu}(-t,-\vec{x}), \qquad \tilde{B}_{a\mu}(t,\vec{x}) = (-1)^{\eta_\mu}(B_{a\mu}(-t,\mathsf{P}\vec{x}) + i\beta\,\partial_t B_{r\mu}(-t,\mathsf{P}\vec{x})) . \tag{37}$$

For a detailed study of the hydrodynamics of $U(1)$ currents using this framework, with and without energy conservation, we remind the reader to [51, 52], where it was shown that generically one finds ordinary diffusive scaling. In the presence of quantum anomalies, the system displays KPZ or coupled-KPZ scaling in one spatial dimension.

## 3.2 The action of non-Abelian hydrodynamics

In this Section we shall construct the action of hydrodynamics for a conserved current associated to a generic non-Abelian (continuous) flavor symmetry $G$. Let us consider coupling the system to an external background gauge field $A_\mu$ taking values in the algebra. Under a gauge transformation,

$$A_\mu \to VA_\mu V^{-1} + iV\,\partial_\mu V^{-1} , \tag{38}$$

where $V(t,\vec{x})$ is an element of $G$. For a theory of a conserved $G$-current the combination (28) generalizes to

$$B_\mu \equiv UA_\mu U^{-1} + iU\partial_\mu U^{-1} , \tag{39}$$

where $U(t,\vec{x})$ is an element of $G$ transforming as

$$U \to UV^{-1} . \tag{40}$$

Note that $B_\mu$ is *invariant* under transformation (38) accompanied with (40). To describe the hydrodynamics of such systems, we consider the Schwinger-Keldysh action where fields are doubled, $B_{1\mu}, B_{2\mu}$, with

$$B_{1\mu} \equiv U_1 A_{1\mu} U_1^{-1} + iU_1\partial_\mu U_1^{-1}, \quad B_{2\mu} \equiv U_2 A_{2\mu} U_2^{-1} + iU_2\partial_\mu U_2^{-1} , \tag{41}$$

which are separately invariant under transformations $V_1, V_2$ defined in (38),(40). Moreover, we impose the diagonal shift symmetry

$$U_1 \to \Lambda U_1, \qquad U_2 \to \Lambda U_2 , \tag{42}$$

where $\Lambda(\vec{x})$ is an element of $G$ that depends arbitrarily on space and is time-independent. Under this transformation, $B_{10}$ and $B_{20}$ are invariant, while

$$B_{1i} \to \Lambda B_{1i}\Lambda^{-1} + i\Lambda\partial_i\Lambda^{-1}, \quad B_{2i} \to \Lambda B_{2i}\Lambda^{-1} + i\Lambda\partial_i\Lambda^{-1} . \tag{43}$$

It is now convenient to introduce the variables

$$B_{r\mu} = \frac{1}{2}(B_{1\mu} + B_{2\mu}), \qquad B_{a\mu} = B_{1\mu} - B_{2\mu} , \tag{44}$$

in terms of which the symmetry principle (43) reads

$$B_{rt} \to \Lambda B_{rt} \Lambda^{-1} \qquad B_{a\mu} \to \Lambda B_{a\mu} \Lambda^{-1} \qquad B_{ri} \to \Lambda B_{ri} \Lambda^{-1} + i\Lambda \partial_i \Lambda^{-1} , \tag{45}$$

i.e. $B_{rt}$ and $B_{a\mu}$ transform in the adjoint, while $B_{ri}$ transforms as a connection and can therefore be used to construct a covariant derivative $\nabla_i = \partial_i - i[B_{ri}, \cdot]$. For example

$$\nabla_i B_{rt} \equiv \partial_i B_{rt} - i[B_{ri}, B_{rt}] , \tag{46}$$

transforms covariantly as $\nabla_i B_{rt} \to \Lambda \nabla_i B_{rt} \Lambda^{-1}$. This will be a convenient building block to write effective actions.

We shall neglect quantum effects of the underlying microscopic system. To this aim, we re-instate powers of $\hbar$ and take $\hbar \to 0$ as we did around eq. (35). Expanding to first order in $\hbar$ (we drop the subscript $r$ from now on),

$$A_{1\mu} = A_\mu + \frac{\hbar}{2}A_{a\mu}, \qquad A_{2\mu} = A_\mu - \frac{\hbar}{2}A_{a\mu}, \qquad U_1 = U\left(1 + i\frac{\hbar}{2}\varphi_a\right), \qquad U_2 = U\left(1 - i\frac{\hbar}{2}\varphi_a\right), \tag{47}$$

where $\varphi_a(t, \vec{x})$ takes values in the algebra of $G$. Then, writing $B_{1\mu} = B_\mu + \frac{\hbar}{2}B_{a\mu}, B_{2\mu} = B_\mu - \frac{\hbar}{2}B_{a\mu}$, we find

$$B_\mu = UA_\mu U^{-1} + iU\partial_\mu U^{-1}, \qquad B_{a\mu} = U(D_\mu \varphi_a + A_{a\mu})U^{-1}, \tag{48}$$

where we defined the covariant derivative with respect to the background $A_\mu$, $D_\mu \psi \equiv \partial_\mu \psi - i[A_\mu, \psi]$, with $\psi$ is in the adjoint representation of $G$.

Let us now look at the dynamical KMS symmetry. As in the previous Section we shall assume that the underlying microscopic system is invariant under $\mathsf{PT}$. Let us take the initial state to be locally a Gibbs ensemble of the form $e^{-\beta(H-\mu_0^A Q^A)}/\text{tr}(e^{-\beta(H-\mu_0^A Q^A)})$, where $\mu_0^A = \mu_0^A(\vec{x})$ is the initial chemical potential, taken to be a slowly-varying function of $\vec{x}$, and $Q^A$ are the generators of the algebra of $G$. As shown in Appendix C, this leads to the symmetry

$$\tilde{B}_{1\mu}(t, \vec{x}) = (-1)^{\eta_\mu} B_{1\mu}(-t, \mathsf{P}\vec{x}), \qquad \tilde{B}_{2\mu}(t, \vec{x}) = (-1)^{\eta_\mu} B_{2\mu}(-t - i\hbar\beta, \mathsf{P}\vec{x}) , \tag{49}$$

where we made explicit the dependence on $\hbar$. The $\mathsf{PT}$ eigenvalue $(-1)^{\eta_\mu}$ of $B_\mu$ is determined by the $\mathsf{PT}$ eigenvalue of the charge density $n$. For example, for the SU(2) spin density $n = n^A \sigma^A$, $n$ flips sign under the composition $\mathsf{PT}$ due to (10). The classical limit $\hbar \to 0$ of (49) is:

$$\tilde{B}_\mu(t, \vec{x}) = (-1)^{\eta_\mu} B_\mu(-t, \mathsf{P}\vec{x}), \qquad \tilde{B}_{a\mu}(t, \vec{x}) = (-1)^{\eta_\mu}(B_{a\mu}(-t, \mathsf{P}\vec{x}) + i\beta \partial_t B_\mu(-t, \mathsf{P}\vec{x})) . \tag{50}$$

We will impose invariance of the action under this transformation in order to encode the presence of local equilibrium.

We now proceed to write down the hydrodynamic action. The combinations $B_\mu$ and $B_{a\mu}$ as given in (48) will be our building blocks. We will write various terms compatible with symmetries (45) and (50), and with the unitarity conditions (29). As in the usual spirit of hydrodynamics, we will follow an expansion in derivatives, and in addition we will perform an expansion in the amplitude of $B_{a\mu}$, as this corresponds to noise corrections. Because of the first condition in (29), each term should be at least liner in $B_{at}$ or $B_{ai}$. At zeroth order in derivatives, there is only one term proportional to $B_{at}$:[7]

$$S_{\text{eff}} = \int d^d x \, dt \, \text{tr}(B_{at} n(B_t)) , \tag{51}$$

---

[7]We choose to take the trace in the adjoint representation; other representations simply lead to different normalizations of the trace which can be re-absorbed in the couplings.

where $n(B_t)$ is an arbitrary function of $B_t$ with values in the Lie algebra of $G$. Although our current discussion applies to a generic non-abelian group $G$, in what follows we shall specialize for concreteness to $G =$ SU(2). In this case, $n$ is an arbitrary odd function of $B_t$ (as $\text{tr}((B_t)^{2n}B_{at}) = 0$). To impose dynamical KMS invariance of the action (51), we require $\text{tr}(B_{at}n(B_t)) \rightarrow \text{tr}((-1)^{\eta_t}(B_{at} + i\beta\partial_t B_t)n((-1)^{\eta_t}B_t)) = \text{tr}((-1)^{\eta_t}B_{at}n((-1)^{\eta_t}B_t)) + i\beta\partial_t\text{tr}(F((-1)^{\eta_t}B_t))$, where $F$ is a suitable function $F$, so that the second term only contributes through a total derivative. Depending on the value of $\eta_t$, $n(B_t)$ may be constrained to be odd independently of $G$.

At first order in derivatives and linear order in $B_{a\mu}$ there can be several terms, depending on the polynomial invariants of the Lie algebra of $G$. For SU(2), there are six terms:

$$-\sigma\,\text{tr}(B_{ai}\partial_t B_i) \qquad -i\,\text{tr}(\lambda B_{ai}[B_t,\partial_t B_i]) \qquad -\lambda'\,\text{tr}(B_{ai}B_t)\,\text{tr}(B_t\partial_t B_i)$$
$$\lambda_1\,\text{tr}(B_{ai}B_t)\,\text{tr}(B_t\nabla_i B_t) \qquad \lambda_2\,\text{tr}(B_{ai}\nabla_i B_t) \qquad i\lambda_3\,\text{tr}(B_{ai}[B_t,\nabla_i B_t])\,, \tag{52}$$

where $\sigma, \lambda, \lambda', \lambda_{1,2,3}$ are arbitrary functions of $\text{tr}(B_t^2)$. In principle we could also include terms proportional to the time component $B_{at}$ which contain derivatives or two or more powers of $B_{a\mu}$, but these can be removed by a suitable redefinition of the hydrodynamic fields [44] There are two terms that contains no derivatives and two powers of $B_{ai}$:

$$i\tilde{\sigma}\,\text{tr}(B_{ai}^2) \qquad i\tilde{\lambda}(\text{tr}(B_{ai}B_t))^2\,, \tag{53}$$

where $\tilde{\sigma}, \tilde{\lambda}$ are functions of $\text{tr}(B_t^2)$, and the factor of i is required by unitarity, see the second eq. in (29). The third eq. in (29) demands $\tilde{\sigma}, \tilde{\lambda} \geq 0$. Imposing dynamical KMS invariance gives $\tilde{\sigma} = \sigma/\beta \geq 0$, $\tilde{\lambda} = \lambda'/\beta \geq 0$, $\lambda_{1,2,3} = 0$, and no condition on $\lambda$. In summary, the most general action up to the first subleading orders in derivatives and $B_{a\mu}$ for SU(2) hydrodynamics is $S_{\text{eff}} = \int d^d x\, dt\, \mathcal{L}$, where

$$\mathcal{L} = \text{tr}\left(B_{at}n - \sigma B_{ai}\partial_t B_i - i\lambda B_{ai}[B_t,\partial_t B_i] - \lambda'\,\text{tr}(B_{ai}B_t)\,\text{tr}(B_t\partial_t B_i) + i\frac{\sigma}{\beta}B_{ai}^2 + i\frac{\lambda'}{\beta}(\text{tr}(B_{ai}B_t))^2\right)\,. \tag{54}$$

From (36), the mean-field hydrodynamic current is $J^\mu = \left.\frac{\delta S}{\delta A_{a\mu}}\right|_{B_{a\mu}=0}$. This gives the charge density $J^t = U^{-1}n(B_t)U = n(\mu)$, where we defined the SU(2) chemical potential

$$\mu \equiv U^{-1}B_t U = A_t - iU^{-1}\partial_t U\,. \tag{55}$$

The spatial component of the current is

$$\begin{aligned} J^i &= U^{-1}(-\sigma\partial_t B_i - i\lambda[B_t,\partial_t B_i] - \lambda' B_t\,\text{tr}(B_t\partial_t B_i))U \\ &= \sigma(E_i - D_i\mu) + i\lambda[\mu,(E_i - D_i\mu)] + \lambda'\mu\,\text{tr}(\mu(E_i - D_i\mu))\,, \end{aligned} \tag{56}$$

where $D_i\mu \equiv \partial_i\mu - i[A_\mu,\mu]$ is the SU(2) covariant derivative, and $E_i = F_{it}$ is the SU(2) background electric field, where $F_{\mu\nu} = \partial_\mu A_\nu - \partial_\nu A_\mu - i[A_\mu, A_\nu]$. To obtain (56) we used

$$U^{-1}\partial_t B_i U = U^{-1}(\partial_t B_i - \partial_i B_0 - i[B_0,B_i] + \partial_i B_0 - i[B_i,B_0])U = -E_i + D_i\mu\,, \tag{57}$$

and

$$\partial_\mu B_\nu - \partial_\nu B_\mu - i[B_\mu,B_\nu] = UF_{\mu\nu}U^{-1}\,, \quad \partial_\nu(U\mu U^{-1}) - i[B_\nu, U\mu U^{-1}] = UD_\nu\mu U^{-1}\,. \tag{58}$$

Setting the SU(2) electric field to zero, and writing in components $\mu = \mu^A\sigma^A$, with $A = 1,2,3$, the current reads

$$J_A^i = -\sigma\partial_i\mu^A + 2\lambda\varepsilon^{ABC}\mu^B\partial_i\mu^C - \lambda'\mu^A\mu^B\partial_i\mu^B\,, \tag{59}$$

which agrees with previous literature [8]. The first term in (59) corresponds to Fick's law for SU(2), and encodes the diffusive behavior of the spin two-point function at long time. The second term is non-dissipative, indeed it has no positivity constraints as discussed below (53); its effect is to rotate the current away from the direction of the SU(2) density, and will play a crucial role in the discussion of Sec. 6.4. We notice that if this term were the only nonvanishing one, i.e. $\sigma, \lambda' = 0$, the resulting dynamics would be integrable in one spatial dimension [53, 54]. The third term can be viewed as an enhancement of diffusion in the direction of the SU(2) density. The advantage of the approach discussed here is two-fold: on one hand, (54) is obtained solely through symmetry principles and, on the other, it provides a systematic approach to evaluate the effect of hydrodynamic fluctuations.

The absence of dynamical KMS symmetry (e.g. for the Floquet-type spin chains studied in Sec. 6 where energy is not conserved) will only lead to minor modifications of our discussion. The terms in (52)-(53) are all allowed in the action (54) as they are consistent with constraints from unitarity (29), and their coefficients can have unrelated values. This will not lead to qualitative changes in the predictions discussed below.

Next, we would like to explicitly show that the frame rotation discussed in Sec. 2.3 leaves the effective action invariant, and to confirm the assumption that $J^i$ transforms covariantly under this frame rotation. The latter transformation can be viewed as a particular gauge transformation acting on $U$ and $A_\mu$ of the form (38) and (40):

$$U \to U e^{i\tilde{\mathcal{U}}t}, \qquad A_\mu \to e^{-i\tilde{\mathcal{U}}t} A_\mu e^{i\tilde{\mathcal{U}}t} + i e^{-i\tilde{\mathcal{U}}t} \partial_\mu e^{i\tilde{\mathcal{U}}t} , \tag{60}$$

where $\tilde{\mathcal{U}}$ is an element of the algebra of SU(2). Since this is a gauge transformation, it will leave $B_\mu$ and $B_{a\mu}$ invariant. Using $\mu = A_t - i U^{-1} \partial_t U$ we see that $\mu \to e^{-i\tilde{\mathcal{U}}t} \mu e^{i\tilde{\mathcal{U}}t}$, and $n$ undergoes the same transformation, hence recovering (20) upon identifying $\tilde{\mathcal{U}} = \frac{1}{2} \mu_{ext}^A \sigma^A$. The shift of the background gauge field $A_t$ precisely accounts for the shift in the background chemical potential in going from microcanonical to (grand) canonical ensemble. Finally, from $J^i = \frac{\delta S}{\delta A_{1i}}\Big|_{A_{a\mu}, \varphi_a = 0}$ we immediately imply that $J^i \to e^{-i\tilde{\mathcal{U}}t} J^i e^{i\tilde{\mathcal{U}}t}$, as stated in Sec. 2.3.

Finally, we emphasize that for larger symmetry groups than SU(2), the actions above can be a bit more complicated, since for example the term $\text{tr}(B_{ai}\{B_t, \partial_t B_i\})$ is no longer vanishing. However, besides adding more nonlinear corrections to the hydrodynamic equations, such effects will change none of the key phenomenology described below. We also note that whenever $\sigma > 0$, all these nonlinear terms (including the $\lambda$ and $\lambda'$ terms in (59)) become irrelevant in the renormalization group sense. If one considers a finely tuned point where $\sigma = 0$, then it may be possible to discover exotic new kinds of hydrodynamics – however such theories will clearly be unstable fixed points.

### 3.3 Adding energy conservation

In the action formulation of hydrodynamics, energy conservation is described in terms of the time reparametrization mode $t \to \sigma(t, \vec{x})$. Here, $t$ denotes the "physical" time, i.e. physical quantities as well as background sources such as $A_\mu$, are given as functions of $t$ (and $\vec{x}$). A nontrivial $\sigma(t, \vec{x})$ induces local dilations of the time scale $dt \to \Lambda dt$, with $\Lambda = \partial_t \sigma(t, \vec{x})$. This in turn corresponds to a local rescaling of the inverse temperature $\beta(t, \vec{x}) \to \partial_t \sigma \beta(t, \vec{x})$, which motivates to identify the local inverse temperature as $\beta = \beta_0 (\partial_t \sigma)^{-1}$, where $\beta_0$ is an overall scale, e.g. the asymptotic value of temperature.[8]

---

[8]For a relativistic system, a more formal way to see this is to couple the system to a metric $g_{\mu\nu}$. For a system in local thermal equilibrium, the inverse temperature at a given point is given by the "time-time" component of the metric, $\beta = \beta_0 \sqrt{-g_{tt}}$. Under time reparametrizations, the metric transforms as $g_{tt} \to g_{tt}(\partial_\sigma t)^2$. Thus, setting to zero the background, $g_{tt} = -1$ the inverse temperature is $\beta = \beta_0(\partial_t \sigma)^{-1}$.

We are interested in Schwinger-Keldysh effective actions, and thus $\sigma$ will be accompanied by the "$a$-variable" $\sigma_a(t,\vec{x})$, responsible for the noise in the energy current. The action will have three symmetries. The first one is space-dependent time shifts:

$$\sigma(t,\vec{x}) \to \sigma(t,\vec{x}) + f(\vec{x}) , \tag{61}$$

where $f(\vec{x})$ is an arbitrary function of $\vec{x}$. This symmetry is motivated by that only the time derivative $\partial_t \sigma$ is a physical quantity, which indeed corresponds to temperature. The second symmetry is constant shifts of $\sigma_a$:

$$\sigma_a(t,\vec{x}) \to \sigma_a(t,\vec{x}) + c , \tag{62}$$

and is a consequence of time-translation invariance. Equivalently, this symmetry is necessary in order to guarantee energy conservation. Finally, we have dynamical KMS invariance [47]

$$\sigma(t,\vec{x}) \to -\sigma(-t,P\vec{x}), \qquad \sigma_a \to -\sigma_a(-t,P\vec{x}) - i\beta(-t,P\vec{x}) , \tag{63}$$

where we assumed that the microscopic system is invariant under PT, as in eq. (50). Having now a local inverse temperature that depends on space and time $\beta(t,\vec{x})$, dynamical KMS invariance for the fields in (50) is modified to

$$\tilde{B}_\mu(t,\vec{x}) = (-1)^\eta B_\mu(-t,P\vec{x}), \qquad \tilde{B}_{a\mu}(t,\vec{x}) = (-1)^\eta(B_{a\mu}(-t,P\vec{x}) + i\beta \hat{V}_\mu(-t,P\vec{x})) , \tag{64}$$

where $\hat{V}_\mu \equiv \beta^{-1}\mathcal{L}_\beta B_\mu = \partial_t B_\mu + \beta^{-1}B_t\partial_\mu\beta$ is, up to a factor of $\beta^{-1}$, the Lie derivative with respect to the space-time vector $\beta\delta^\mu_t$ [44].

To include energy conservation in the hydrodynamic action (54) we use expansion in derivative and in the amplitudes of $B_{a\mu}$ and $\sigma_a$, leading to

$$S = \int dt d^d x \Bigg\{ -\varepsilon\partial_t\sigma_a - (\kappa\partial_i\beta + \alpha\,\mathrm{tr}(B_t\hat{V}_i))\partial_i\sigma_a + i\kappa(\partial_i\sigma_a)^2 +$$
$$\mathrm{tr}\Bigg( B_{at}n - \sigma B_{ai}\hat{V}_i - i\lambda B_{ai}[B_t,\partial_t B_i] - \lambda' B_{ai}B_t\,\mathrm{tr}(B_t\hat{V}_i) - \alpha\beta^{-1}B_{ai}B_t\partial_i\beta \tag{65}$$
$$+ i\frac{\sigma}{\beta}B_{ai}^2 + i\frac{\lambda'}{\beta}(\mathrm{tr}(B_{ai}B_t))^2 + 2i\frac{\alpha}{\beta}B_t B_{ai}\partial_i\sigma_a \Bigg) \Bigg\} ,$$

where $n,\sigma,\lambda,\lambda',\varepsilon,\kappa,\alpha$ are arbitrary functions of $\mathrm{tr}(B_t^2)$ and of $\beta$. Various terms are proportional to the combination $\hat{V}_i$ instead of just $\partial_t B_i$ as a consequence of dynamical KMS invariance, as one can verify. The latter invariance also implies that $\varepsilon$ and $n$ satisfy the thermodynamic relations

$$\frac{\partial(\beta p)}{\partial\beta} = -\varepsilon, \qquad \frac{\partial(\beta p)}{\partial(\beta B_t)} = n , \tag{66}$$

where $p$ is an arbitrary function of $\mathrm{tr}(B_t^2)$ and $\beta$. Conjugating the second equation by $U$ and using $U^{-1}B_t U = \mu$ we recover the thermodynamic relation $d(\beta p) = -\varepsilon d\beta + n d(\beta\mu)$, which rearranges into the Gibbs-Duhem equation with a non-Abelian chemical potential

$$dp = s d(\beta^{-1}) + \mathrm{tr}(n d\mu) , \tag{67}$$

where the entropy density is $s = \beta(p + \varepsilon - \mathrm{tr}(\mu n))$. In (65) we omitted terms proportional to $B_{at}^2, (\partial_t\sigma_a)^2, B_{at}\partial_t\sigma_a$ because we can fix them to zero up to a redefinition of $\mu,\beta$ [44].

Since energy conservation is obtained by varying the Lagrangian with respect to $\sigma_a$, the energy current can be read off directly by looking at terms proportional to $\partial_t\sigma_a$ and $\partial_i\sigma_a$, giving

$$J_\varepsilon^t = \varepsilon, \qquad J_\varepsilon^i = \kappa\partial_i\beta + \alpha\,\mathrm{tr}(\mu V_i) , \tag{68}$$

where $V_i \equiv U^{-1}\hat{V}_i U = -E_i + D_i\mu + \beta^{-1}\mu\partial_i\beta = -E_i + \beta^{-1}D_i\hat{\mu}$, where we used (57), and $\hat{\mu} \equiv \beta\mu$. The SU(2) current is

$$J^t = n, \qquad J^i = -\sigma V_i - \mathrm{i}\lambda[\mu, V_i] - \lambda'\mu\,\mathrm{tr}(\mu V_i) - \alpha\mu\beta^{-1}\partial_i\beta \;. \tag{69}$$

Let us now expand around a background temperature and chemical potential: $\beta = \beta_0 + \delta\beta$ and $\hat{\mu} = \hat{\mu}_0 + \delta\hat{\mu}$ and set the SU(2) background $A_\mu$ to zero, where $\beta_0, \hat{\mu}_0$ denote the background values. Choosing $\hat{\mu}_0 = \hat{\mu}_0^A\sigma^A$ with $\hat{\mu}_0^A = \bar{\mu}_0\delta^{A3}$, with $\delta^{AB}$ denoting the components of the identity in flavor space, and decomposing $\delta\hat{\mu} = (\delta\hat{\mu}^I, \delta\hat{\mu}^3)$, with $I = 1, 2$, gives

$$J_\varepsilon^i = \kappa\partial_i\delta\beta + \alpha\beta_0^{-2}\bar{\mu}_0\partial_i\delta\hat{\mu}^3, \qquad J_3^i = -\sigma\beta_0^{-1}\partial_i\delta\hat{\mu}^3 - \lambda'\beta_0^{-3}\bar{\mu}_0^2\partial_i\delta\hat{\mu}^3 - \alpha\beta_0^{-2}\bar{\mu}_0\partial_i\delta\beta$$
$$J_I^i = -\sigma\beta_0^{-1}\partial_i\delta\hat{\mu}^I - 2\lambda\beta_0^{-2}\hat{\mu}_0\varepsilon^{IJ}\partial_i\delta\hat{\mu}^J \;. \tag{70}$$

We also have $\varepsilon = \varepsilon_0 + \delta\varepsilon$ and $n = n_0 + \delta n$, with

$$\delta\varepsilon = \partial_\beta\varepsilon\delta\beta + \partial_{\hat{\mu}^3}\varepsilon\delta\hat{\mu}^3, \qquad \delta n^3 = -\partial_{\hat{\mu}^3}\varepsilon\delta\beta + \chi_3\delta\hat{\mu}^3, \qquad \delta n^I = \chi_0\delta\hat{\mu}^I \;, \tag{71}$$

where $\chi_3 = \partial_{\hat{\mu}^3}n^3$ and $\chi_0 = \frac{1}{2}\partial_{\hat{\mu}^I}n^I$ are SU(2) susceptibilities. In (71) we used that, from (66), $-\frac{\partial\varepsilon}{\partial\hat{\mu}^A} = \frac{\partial n^A}{\partial\beta}$, and $n_1$ is defined through $\partial_{\hat{\mu}^I}n^J = n_1\delta_I^J$. The conservation equations then split into a scalar and a vector sector with respect to the U(1) transformations preserved by $\hat{\mu}_0$:

$$\partial_t\begin{pmatrix}\delta\varepsilon \\ \delta n^3\end{pmatrix} = \begin{pmatrix}\kappa & \alpha\beta_0^{-2}\bar{\mu}_0 \\ \alpha\beta_0^{-2}\bar{\mu}_0 & \sigma\beta_0^{-1} + \lambda'\beta_0^{-3}\bar{\mu}_0^2\end{pmatrix}\begin{pmatrix}-\partial_\beta\varepsilon & -\partial_{\hat{\mu}^3}\varepsilon \\ -\partial_{\hat{\mu}^3}\varepsilon & \chi_3\end{pmatrix}^{-1}\partial_i^2\begin{pmatrix}\delta\varepsilon \\ \delta n^3\end{pmatrix}, \tag{72}$$

$$\partial_t\delta n^I = \chi_0^{-1}\beta_0^{-1}(\sigma\delta^{IJ} + 2\lambda\beta_0^{-1}\bar{\mu}_0\varepsilon^{IJ})\partial_i^2\delta n^J \;. \tag{73}$$

The scalar sector contains two diffusive modes carried by linear combinations of $\delta\varepsilon$ and $\delta n^3$, as in the standard hydrodynamics of U(1) charge and energy conservation. The vector sector is carried by $\delta n^I$ and has two modes with diffusive and magnon contributions:

$$\omega = -\mathrm{i}Dk^2 \pm Ak^2 \;, \tag{74}$$

with $D = \sigma/(\beta_0\chi_0)$ and $A = 2\bar{\mu}_0\lambda/(\beta_0^2\chi_0)$ [8,55]. See also the recent discussion in [56].

We shall now perform a scale analysis. The dispersion relations just found imply the scaling $\omega \sim k^2$. Requiring that the first and the fourth terms in (65) be dimensionless gives $\sigma_a, \varepsilon \sim k^{d/2}$. Similarly, one infers that $\varphi_a, n \sim k^{d/2}$. This is in fact the standard scaling of diffusive conserved densities and their associated noise variables. It is then easy to see that the nonlinear couplings in (65) have positive momentum dimension: the terms proportional to $\sigma_1, \sigma_3$ scale like $k^{d/2}$, while the terms proportional to $\sigma_2$ scale like $k^d$. This means that all the nonlinearities are irrelevant in the renormalization group sense, and thus the dispersion relation (74) is not affected by hydrodynamic fluctuations at $O(k^2)$. As a consequence, the diffusion constant $D$ in (74) remains finite at late time, in disagreement with the logarithmic enhancement of eq. (1). We then see that, as far as (fluctuating) hydrodynamics is concerned, only the presence of additional charges can potentially lead to a logarithmic enhancement of diffusion at very late times. We will discuss this scenario in Section 5.

The nonlinearities in (65) still lead to high-temperature non-analyticities in transport. In particular, the term proportional to $\lambda$ in Eq. (59) can be shown to give a non-analytic contribution to the conductivity of the form

$$\sigma(\omega) \sim \sigma^{(0)} + \lambda^2|\omega|^{d/2} + \cdots \;, \tag{75}$$

as can be obtained from a one-loop calculation of the fluctuating hydrodynamic path integral of the action (65) (or (54)). This same singularity was found in Ref. [57], where high-temperature non-analyticities in transport of many-body chains were first discovered. In that

case energy conservation was crucial in order to have this effect. Here, the nonlinearity associated to $\lambda$ is present even in systems which break energy conservation; the crucial ingredient is the presence of multiple densities (as was clear also from [57]), even without energy conservation. In Section 6.4 we will show that the above nonlinearities are still crucial to explain the puzzle related to eq. (1).

# 4 Fluids with other non-Abelian symmetries

The purpose of this Section is to explain why non-Abelian flavor symmetries have a very different impact on hydrodynamics than other non-Abelian symmetries in hydrodynamics. A classic example of such a symmetry is rotational invariance: the generators of rotation do not commute with momentum, so the symmetry algebra of an ordinary liquid is non-Abelian. A more exotic example is a fluid with multipole constraints. In each case, the non-Abelian symmetry group is intimately tied to a mixture of spacetime symmetry with an Abelian flavor symmetry. We will show how that qualitatively changes the effective theory.

## 4.1 Fractons

We first describe fluids with a conserved U(1) charge *and* dipole moment, whose hydrodynamics was recently formulated in [58]; see also [59–62]. An instructive cartoon is to start by supposing that there is a local conserved density $\rho$ corresponding to charge, and $s_i$ corresponding to local dipole density *orthogonal* to $x_i\rho$: namely, the total conserved dipole moment can be written as

$$P_i := \int d^d x \ (x_i \rho + s_i). \tag{76}$$

$x_i$ represents the space-only coordinates in standard Einstein index notation. To understand this prescription, it can be helpful to imagine the explicit coarse-graining prescription of hydrodynamics. We divide up space into boxes of length $L$. For simplicity, let us take $d = 1$. Charge and dipole densities are

$$n(x) = \frac{1}{L} \int\limits_{x-L/2}^{x+L/2} dy \, \rho(y), \qquad d(x) = \frac{1}{L} \int\limits_{x-L/2}^{x+L/2} dy \, y\rho(y) \qquad s(x) = d(x) - xn(x). \tag{77}$$

Assuming the system thermalizes, after a sufficiently long time the two-point function of $\rho$, $G(x,t) = \langle \rho(t,x)\rho(0,0) \rangle$, will be a slow function of $x$ compared to the scale $L$. The two-point function of the coarse-grained density $n(x)$ in this regime can then be approximated by $G$, i.e.

$$\langle n_x(t)n_0 \rangle = \frac{1}{L^2} \int\limits_{x-L/2}^{x+L/2} dy \int\limits_{-L/2}^{L/2} dz \, G(y-z,t) \to G(x,t). \tag{78}$$

Similarly, the two-point function of the coarse-grained dipole density $d_x$ becomes:

$$\langle d_x(t)d_0 \rangle = \frac{1}{L^2} \int\limits_{x-L/2}^{x+L/2} dy \int\limits_{-L/2}^{L/2} dz \, yz\, G(y-z,t) \to -\frac{L^2}{12} x\partial_x G(x,t), \tag{79}$$

where we used that, taking $x \gg L$, $\int_{-L/2}^{L/2} \frac{dz}{a} z G(y-z,t) \approx -\frac{L^3}{12}\partial_y G(y,t)$ via a Taylor expansion in $z$.

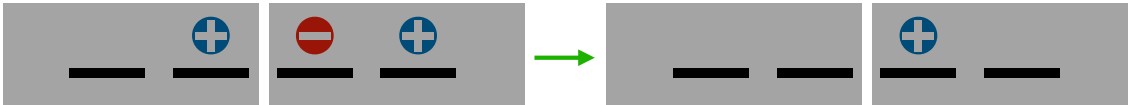

Figure 1: Allowed dynamics in a one-dimensional dipole-conserving lattice model. Gray shaded regions denote different coarse grained blocks as in (77). This move shows that $\int \mathrm{d}x s(x)$ is not conserved in general. Only charge density represents a generic hydrodynamic degree of freedom.

We see that the two-point function of $d(x)$ is entirely determined by that of $n(x)$. In the long time limit, charge density tends to vary over long scales, and thus the dipole charge is effectively carried by the coarse-grained charge density.

One might ask whether the argument above is too fast – perhaps on a discrete lattice (with sites labeled by integers), for example, the function $G(x, t)$ oscillates rapidly every other $x$, thus encoding additional structure that is not captured by the hydrodynamic mode $\rho$ in the continuum effective theory. Can it be possible that this extra structure can encode a new conservation law for $s(x)$, the microscopic dipole density? As shown in Fig. 1, this possibility can be generically ruled out. The reason is that even in a local region of the lattice, the local dipole density defined through the coarse graining procedure (77) can decay through the motion of a local charge surplus in space: (76) permits $s$ to decay via this channel as only the total dipole moment is conserved. At the level of the equations of motion, one will generically find a non-hydrodynamic equation of motion for this microscopic dipole density:

$$\partial_t s = -\frac{s - \partial_x n}{\tau} + D_2 \partial_x^2 s + \cdots. \tag{80}$$

The key difference then, between this dipole-conserving theory and the non-Abelian flavor hydrodynamics of Section 3 is that the motion of flavor charges through space does *not* allow other flavor charges to decay away. The mixing of spacetime symmetries with charge conservation does not lead to new degrees of freedom, but rather adds *additional constraints*. For example, in the dynamical process shown in Fig. 1, the decay of $s(x)$ is sensitive to the precise location of the coarse-graining. So we conclude that in the hydrodynamic equations for both $\partial_t s$ and $\partial_t n$, the right hand sides can depend only on the combination $s - \partial_x n$, or higher derivative corrections $\partial_x s$, $\partial_x^2 n$, etc. Combining (80) with

$$\partial_t n = D_1 \partial_x (\partial_x n - s) - D_3 \partial_x^4 n - D_4 \partial_x^3 s + \cdots \tag{81}$$

we obtain a fourth-order subdiffusive equation for $n$:

$$\partial_t n = -\left(D_4 + D_3 + D_1 D_2 \tau\right) \partial_x^4 n + \cdots \tag{82}$$

in agreement with the general framework of [58]; see also similar arguments to the above paragraph in [63] in the context of an experiment in a cold atomic gas in a tilted optical lattice.

Indeed, when the commutator of total momentum $\mathcal{P}_x$ with a charge (such as dipole density $P_x$) gives another charge (such as $Q$), i.e.[9]

$$[\mathcal{P}_x, P_x] = \mathrm{i}Q, \tag{83}$$

then only $Q$ is a hydrodynamic mode. This captures both the multipole symmetry case discussed above and the rotation symmetry case that we will discuss next. In the EFT formalism

---

[9]The non-commutativity of multipole algebras in the presence of momentum was first appreciated in [64].

we have introduced above, the essential idea is as follows. Let $\varphi_x$ and $\varphi$ be local phase degrees of freedom, as in Section 3, for the dipole and charge respectively. Then (neglecting background gauge fields) the invariant quantities are $\partial_t \varphi_a$, $\partial_t \varphi_{xa}$, $\partial_t \varphi$, $\partial_t \varphi_x$, $\partial_x \varphi_{xa}$, and $\partial_x \varphi_a - \varphi_{xa}$. The latter invariant mixes together the two symmetries in an important way. The most general effective action at linear order in $a$-fields then takes the form

$$S = \int dt\, dx\, (\rho \partial_t \varphi_a + \rho_x \partial_t \varphi_{xa} + J_x(\partial_x \varphi_a - \varphi_{xa}) + J_{xx} \partial_x \varphi_{xa}), \tag{84}$$

where $\rho$, $\rho_x$ can depend on $\partial_t \varphi$ and $\partial_t \varphi_x$, then we can see that in the hydrodynamic limit, the second term is negligible as it is higher derivative than the third term. The $\varphi_{xa}$ equation of motion gives

$$J_x = \partial_x J_{xx}, \tag{85}$$

precisely as claimed in [58] – the dipole current $J_{xx}$ is the fundamental hydrodynamic operator, whose derivative gives the ordinary charge current.

## 4.2 Rotational invariance

Essentially identical arguments to the above allow us to argue that a rotationally invariant fluid with both momentum and angular momentum conservation will not have a new degree of freedom corresponding to angular momentum, but instead will obey extra constraints in the hydrodynamic regime [65]; see also [66]. Let $p_i(\vec{x})$ denote the momentum density in $d$ dimensions, and let $L_{ij}(\vec{x})$ denote a local angular momentum density, where only the total angular momentum

$$J_{ij} = \int d^d x \left( x_i p_j - x_j p_i + L_{ij} \right) \tag{86}$$

is conserved. Repeating the argument of the previous subsection implies that the equation of motion for $L_{ij}$ takes the schematic form

$$\partial_t L_{ij} = -\frac{L_{ij} - (\partial_i p_j - \partial_j p_i)}{\tau} + \cdots \tag{87}$$

and that the equation of motion for momentum takes the form

$$\partial_t p_i = -\partial_j \tau_{ij}, \qquad \tau_{ij} = \tau_{ij}[L_{mn} - (\partial_m p_n - \partial_n p_m), \partial_m p_n + \partial_n p_m, \partial_m L_{np}, \partial_m \partial_n p_p, \ldots]. \tag{88}$$

In an ordinary fluid, there would also be charge/mass and energy conservation included in ... as well. The antisymmetric contribution to the stress tensor $\tau_{ij}$ is, at leading order in the hydrodynamic limit, proportional to $L_{ij} - \partial_i p_j + \partial_j p_i$, which vanishes according to (87). Therefore, at leading order in hydrodynamics, the stress tensor must be symmetric, even when we include an explicit degree of freedom representing internal rotational dynamics.

## 4.3 No non-Abelian fracton hydrodynamics

Interestingly, we also observed that there is no non-trivial hydrodynamics with propagating degrees of freedom that can be found by considering possible non-Abelian flavor extension of "fracton hydrodynamics." We begin by a consideration of systems in one dimension, where there are no non-trivial possibilities whatsoever, before discussing higher dimensions, where the only non-trivial possibility is diffusion along subdimensional manifolds.

Let us start by assuming that we have a one dimensional lattice model where all flavor charges $Q^A$ are conserved, as is the dipole moment of just one flavor charge $P^1 = \sum_x x Q_x^1$ i.e. $[U(t), Q^A] = 0$ for all $A$, and also $[U(t), P^1] = 0$. We also assume the group $G$ is simple, and

will return to this point later. Now from the Jacobi identify for commutators, it follows that $[Q^A, P^1]$ is also conserved, for any $A$. However $i[Q^A, P^1] = -f^{A1C}P^C$, and thus $P^C$ must also be conserved, for any $C$. Thus, if all flavor charges are conserved and so is the dipole moment of one charge, then the dipole moment of *every* charge must be conserved i.e. $[U(t), P^A] = 0$ for all $P^A$. Now by another application of the Jacobi identity for commutators, it follows that $[U(t), [P^A, P^B]] = 0$ as well. Therefore, the following quantity is also conserved:

$$i[P^A, P^B] = -f^{ABC} \sum_{x \in \Lambda} x^2 Q_x^C. \tag{89}$$

We conclude that all quadrupole moments are also conserved! Obviously, arbitrarily higher moments of conserved flavor charge can also be generated, and so there is no hydrodynamics at any perturbative order in derivatives. To recover subdiffusion, the only flavors with dipole (or higher) moments conserved must correspond to mutually commuting charges. Yet this would, of course, microscopically break the flavor symmetry group. An exception to the statements above arise if the group $G$ is not simple. For example, if $G = U(1) \times U(1) \times \cdots \times G_{\text{non-Ab}}$, it is possible to have multipole conservation laws for the $U(1)$ charges, as they commute with all other charges.

We now show that not only does a 'fractonic' extension of a theory with non-Abelian charges not have a hydrodynamic description, it necessarily has totally trivial dynamics, starting with the case of one spatial dimension. Consider a lattice system and work in the basis of product states in the $Q^A$ charge basis. Any two distinct product states will differ in at least one of their $Q^A$ multipole moments, and since all the multipole moments are conserved, each such state will be in a separate symmetry sector. The time evolution operator will thus be purely diagonal in this basis. This argument would have worked just as well for any $A$, and thus the time evolution operator must be diagonal in *any* local product state basis. The only option is a time evolution operator that acts as the identity on every site - corresponding to totally trivial dynamics.

These arguments manifestly extend to higher spatial dimensions, with all components of dipole conserved. They similarly also rule out non-Abelian theories in higher dimensions with subsystem symmetry along all directions. The easiest way to see this is to note that subsystem symmetry automatically implies dipole conservation (if charge is conserved in every hyperplane orthogonal to $z$ then the $z$ component of dipole is also conserved), and by the above argument conservation of all components of dipole is sufficient to trivialize the dynamics.

Finally, let us consider a higher dimensional system with dipole (and hence multipole) conservation along only one direction, $\hat{x}$. This implies subsystem symmetry in hyperplanes orthogonal to $\hat{x}$ only. This is not sufficient to totally trivialize the dynamics. Following [58], we expect that the conserved densities will be governed by equations of the form $\partial_t \rho = a\nabla_\perp^2 \rho + b\nabla_\perp^2 \partial_x^2 \rho$, where $\nabla_\perp^2$ denotes the Laplacian in the hyperplane orthogonal to $x$, and $a$ and $b$ are numerical constants. At leading order in derivatives, this just corresponds to ordinary diffusion in hyperplanes orthogonal to $\hat{x}$.

## 5 A theory with logarithmically-enhanced diffusion

In thermalizing systems, logarithmically diverging transport parameters at low frequencies are a hallmark of marginally irrelevant hydrodynamic fluctuations [67]. These arise schematically as a hydrodynamic loop contribution to the Kubo formula

$$\delta D \sim \frac{1}{D^{n/2}} \log \frac{1}{\omega}, \tag{90}$$

with $n \in \mathbb{N}$ (a detailed example will be given below). Higher-loop contributions are similarly divergent. The leading logarithmic divergences can be resummed by solving the $\beta$-function equation

$$\beta_D = \frac{\delta D}{\delta \log \frac{1}{\omega}} \sim \frac{1}{D^{n/2}}. \tag{91}$$

The solution leads to anomalous diffusion constants that grow logarithmically with time

$$D(t) \sim \log^\alpha t, \tag{92}$$

with $\alpha = 2/(n+2)$. Marginally irrelevant hydrodynamic fluctuations are common in $d = 2$ spatial dimensions, where the leading divergence (90) arises at one-loop; examples include regular fluid dynamics [67] (where $\alpha = 1/2$), surface chiral metals [52] (where $\alpha = 2/3$) as well as driven-dissipative systems (see e.g. [68, 69]). Logarithmically-enhanced diffusion is also possible in $d = 1$ spatial dimensions, if the leading effect of hydrodynamic fluctuations arises at two-loops – a possibility realized for example in surface growth with reflection symmetry [70]. In this Section we find a hydrodynamic theory involving non-abelian densities in $d = 1$ that has similarly anomalous diffusion. As we show below, this requires emergent symmetries leading to additional slow densities. As we detail in Section 6.3, emergent symmetries are essentially ruled out by our numerical results for a class of Heisenberg chains, and therefore the mechanism presented here is not a viable explanation of the proposed [32] logarithmic anomalies in these systems. However we hope that this mechanism, although unrealistic in this context, can serve as an illustration of logarthmically-enhanced diffusion in $d = 1$.

Hydrodynamic interactions arise from nonlinearities in the constitutive relation for the spin current $J^A \equiv J_x^A$:

$$J^A = -D\nabla n^A + \cdots. \tag{93}$$

Since hydrodynamic densities scale as $n \sim k^{d/2} = k^{1/2}$ (see Sec. 3.3), a nonlinear term in (93) is marginal (i.e. scales like the diffusive term) if it contains three hydrodynamic densities and no gradient. Parity symmetry requires one (or three) of these to be odd under parity. We will assume for simplicity that spin density $n^A$ is the only hydrodynamic variable that is charged under the flavor symmetry[10], and denote the parity-odd density by $\tilde{\pi}$. A relevant nonlinearity $J^A \sim \tilde{\pi}n^A$ would lead to KPZ scaling [67, 71]; it is forbidden if $\tilde{\pi}$ is even under time-reversal. If the theory also contains an emergent parity-even, time-reversal-odd density $\tilde{\varepsilon}$, then the leading nonlinearity in the constitutive relation will be marginal

$$J^A = -D\nabla n^A + \lambda\, n^A \tilde{\pi}\tilde{\varepsilon} + \cdots, \tag{94}$$

with $\lambda \in \mathbb{R}$. Now the pair of densities $\tilde{\pi}$, $\tilde{\varepsilon}$ associated with emergent symmetries will typically form a sound mode, which would suppress loop contributions to the spin diffusivity. We will assume that this does not happen and that $\tilde{\pi}$ and $\tilde{\varepsilon}$ diffuse with constants $D_\epsilon, D_\pi$ – in this sense this situation is fine-tuned. The new term in (94) will lead to a two-loop correction to the spin retarded Green's function (see Appendix B for conventions on correlation functions)

$$G^R_{n^A n^B}(\omega, k) = \frac{\delta_{AB}\chi D k^2}{-i\omega + Dk^2 + \Sigma(\omega, k)} \tag{95}$$

of the form

$$\Sigma(t,x) \sim \langle(\nabla n^A \tilde{\pi}\tilde{\varepsilon})(\nabla n^A \tilde{\pi}\tilde{\varepsilon})\rangle(x,t) \sim \nabla^2 \frac{e^{-\frac{x^2}{2|t|}\left(\frac{1}{D} + \frac{1}{D_\varepsilon} + \frac{1}{D_\pi}\right)}}{\sqrt{DD_\varepsilon D_\pi |t|^3}}. \tag{96}$$

---

[10] Lifting this assumption allows for certain exotic possibilities, e.g: the emergence of a spin-3 parity-odd density could lead to a marginal interaction $J^A \sim q^{ABC} n_B n_C$ (one advantage of this scenario is that a sound mode does not have to be fine tuned away, see main text). The numerics in Sec. 6 however also rules out this possibility in a broad range of frequently studied models.

Fourier transforming leads to a logarithmic correction to the diffusion constant

$$\delta D \sim \lim_{k \to 0} \frac{\Sigma(\omega, k)}{k^2} \sim \int \mathrm{d}x \mathrm{d}t \, \mathrm{e}^{\mathrm{i}\omega t} \frac{\mathrm{e}^{-\frac{x^2}{2|t|}\left(\frac{1}{D} + \frac{1}{D_\varepsilon} + \frac{1}{D_\pi}\right)}}{\sqrt{DD_\varepsilon D_\pi |t|^3}} \sim \frac{\log \frac{1}{\omega}}{\sqrt{DD_\varepsilon + D_\varepsilon D_\pi + D_\pi D}} \,. \tag{97}$$

The diffusion constants associated with the emergent symmetries $D_\varepsilon$, $D_\pi$ will receive similar corrections. These logarithmic contributions can be resummed by solving the (coupled) $\beta$-function equations, as described above. One finds that all diffusion constants behave, as $\omega \to 0$ (and thus $t \to \infty$), as

$$D \sim \log^{1/2} \frac{1}{\omega}, \qquad \text{or} \qquad D \sim \log^{1/2} t \,. \tag{98}$$

At intermediate times, it is possible that $D_\varepsilon$ and $D_\pi$ have not yet reached their asymptotic behavior (98) and are approximately constant. If this case, one finds from (97) that the spin diffusivity behaves as $D \sim \log^{2/3} t$ if $D \gg \min(D_\epsilon, D_\pi)$ and $D \sim \log t$ if $D \ll \min(D_\epsilon, D_\pi)$.

# 6 Spin chains with SU(2) symmetry

In this Section we apply the formalism mentioned above to spin chains with SU(2) symmetry. There appears to be some controversy in the literature as to whether such spin chains, in the generic non-integrable case, should exhibit conventional diffusion [18, 21, 28–30] or superdiffusion [16, 24, 32]. We aim to resolve this controversy.

## 6.1 The classical Heisenberg spin chain

A canonical example of a lattice model with SU(2) flavor symmetry is the classical Heisenberg model with Hamiltonian on an $L$-site lattice in one dimension:

$$H = J \sum_{i=1}^{L-1} \vec{S}_i \cdot \vec{S}_{i+1}, \tag{99}$$

where $|\vec{S}_i| = 1$ is a classically constrained vector. Using the classical Poisson brackets

$$\{S_i^A, S_j^B\}_{\mathrm{PB}} = \epsilon^{ABC} S_i^C \delta_{ij} \tag{100}$$

the equation of motion

$$\frac{\mathrm{d}}{\mathrm{d}t} \vec{S}_i = \{H, \vec{S}_i\}_{\mathrm{PB}} \tag{101}$$

is widely believed to generate chaotic and non-integrable classical dynamics [21]. (In contrast, the quantum spin-$\frac{1}{2}$ model is integrable [72, 73]).

There is an old debate in the literature about the nature of hydrodynamics and spin diffusion in the classical Heisenberg model [16, 18, 21, 24]. This debate has been revived in the recent literature by recent work [32] arguing for a logarithmically enhanced diffusion constant: $D(t) \sim \log^{4/3} t$ as in (1), with other authors arguing for conventional spin diffusion due to the lack of integrability [29, 30].

Let us briefly review the motivation [32] for the logarithmically enhanced spin diffusion constant (1). In the continuum limit, (99) is integrable with an emergent conserved quantity associated to invariance under continuous translations (i.e. momentum):

$$\tau = \frac{\vec{S} \cdot (\partial_x \vec{S} \times \partial_x^2 \vec{S})}{\partial_x \vec{S} \cdot \partial_x \vec{S}}. \tag{102}$$

$\tau$ approximately obeys the classical Burgers equation, which in one dimension leads to Kardar-Parisi-Zhang (KPZ) scaling [74] in the presence of noise, which we denote schematically with $\xi$. Indeed, KPZ scaling can be observed for a brief transient period at early times in the dynamics [32]. However, the authors of [32] argued that at sufficiently late times, the noise spectrum of the Burgers equation weakens algebraically:

$$\overline{\xi(t)\xi(s)} \sim t^{-1/2}\delta(t-s), \tag{103}$$

and this leads [75] to a modification of KPZ scaling compatible with (1), even at infinite temperature.

Two specific possible flaws in this argument are that: (*1*) it appears to rely on a breakdown of ergodicity, due to the time dependence in (103), yet we are most interested in looking at diffusion in equilibrium correlators; (*2*) it has been emphasized in Ref. [37] that the continuum limit of the Heisenberg model does not apply at infinite temperature, as umklapp processes cannot be ignored.

From the perspective of our effective theory, we make two general comments: (*1*) SU(2) symmetry is not sufficient to render $\tau$ a hydrodynamic mode, and even if the effects of the lattice were small, they are expected to be "dangerously irrelevant", breaking conservation laws and qualitatively changing the character of hydrodynamics. (*2*) We can easily test for whether $\tau$ is hydrodynamic on the lattice in numerical simulations. As detailed below, we find no evidence that $\tau$ is long-lived. Moreover, after an exhaustive search, we did not find any evidence for non-trivial structure to the hydrodynamics of the classical Heisenberg chain beyond ordinary spin and energy diffusion. As predicted by our general effective theory, each component of spin obeys, at leading order, a separate diffusion equation.

## 6.2 Numerical results

In this Section, we numerically study the dynamics of several classical models with SU(2) symmetry and compare with the result obtained from effective field theory. The algorithm we are using is the first method described in Appendix D. We first consider the Heisenberg model (99), which is a chaotic system with both energy and spin conserved. These two conservation laws can be reflected in the long time diffusive behavior in the autocorrelation function

$$C_{\mathcal{O}}(t) = \langle O_i(t)O_i(0)\rangle - \langle \mathcal{O}_i\rangle^2, \tag{104}$$

where $\mathcal{O}$ is chosen as $S_i^z$ or the local energy $\epsilon_i = \vec{S}_i \cdot \vec{S}_{i+1}$. We numerically check these two quantities in the Heisenberg model with periodic boundary conditions and setting $J = 1$. The average $\langle\cdots\rangle$ is taken in both initial states and spatial direction. As shown in Fig. 2(a), we notice that $C_\epsilon(t) \sim 1/\sqrt{t}$ at late time, while for $C_z(t)$ a small deviation from $1/\sqrt{t}$ appears to arise. We further plot the numerically extracted diffusion constant

$$D(t) = \frac{1}{tC(t)^2} \tag{105}$$

as a function of $t$ in Fig. 2(b) and we find that while $D_\epsilon(t)$ approaches a constant as time evolves, $D_z(t)$ increases continuously with time, albeit quite slowly. Similar behavior has also been found in Ref. [32], where they propose that $D_z(t) \sim [\log(t)]^{4/3}$. In Fig. 2(c), we carefully plot $[D_z(t)]^{3/4}$ as a function of $\log(t)$ and we find that it is a straight line when $2 < \log t < 4$ and the curve starts to bend down when $\log t > 4$. It is quite possible that in this Heisenberg model, when the time is long enough, $D_z(t)$ will eventually approach a constant.[11] Further calculation at finite $\beta$ (see Fig. 2(c)) indicates that $D_z(t)$ also grows with the time but the saturation appears much faster. Notice that this logarithmic correction does not appear in $C_\epsilon(t)$: See Fig. 2(a) and Fig. 2(d).

---

[11]Numerically it is hard to extract $D(t)$ when $\log t > 7$ since $C(t)$ becomes very small and noisy.

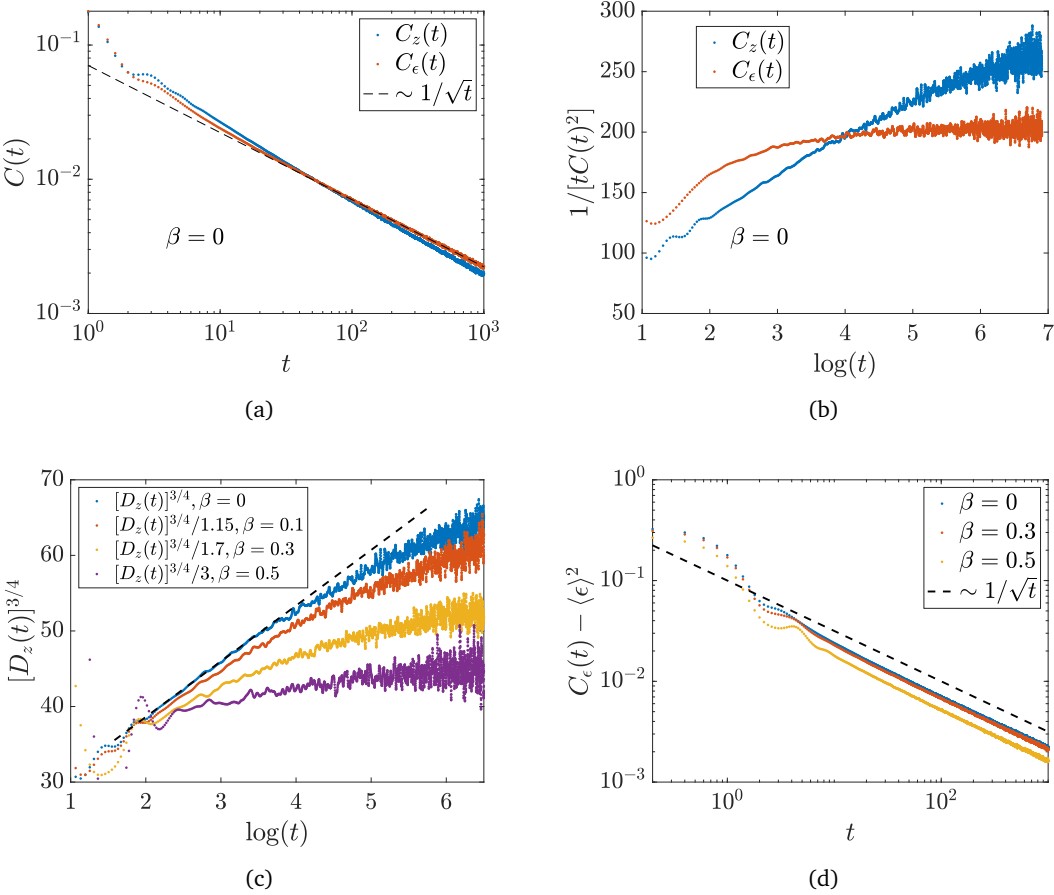

Figure 2: $S^z$ correlator $C_z(t) = \langle S_z(t)S_z(0)\rangle$ and energy density correlator $C_\epsilon(t) = \langle \epsilon(t)\epsilon(0)\rangle$ in the Heisenberg model with $L = 1000$. (a) $C_z(t)$ and $C_\epsilon(t)$ vs $t$ on the log-log scale. (b) Diffusion constant $D(t)$ vs $\log t$. (c) (Rescaled) spin diffusion constant $[D_z(t)]^{3/4}$ vs $\log(t)$ at various temperatures $1/\beta$. (d) Energy correlator vs $t$ at various $\beta$ on the log-log scale.

Let us further explore $D_z(t)$ in a few variants of the Heisenberg model at the same system size. (*1*) We consider the XXZ model

$$H = \sum_i S_i^x S_{i+1}^x + S_i^y S_{i+1}^y + \Delta S_i^z S_{i+1}^z, \qquad (106)$$

with $\Delta \neq 1$. In this case, only the $z$-component of $\vec{S}$ is conserved and, as shown in Fig. 3(a), $D_z(t)$ saturates to a constant after an early time growth. $D_z(t)$ approaches a constant faster as we move away from $\Delta = 1$, in agreement with the numerics of [32]. We also present $D_\epsilon(t)$ in Fig. 3(b) for comparison. (*2*) Let us now add a third nearest neighbor interaction term to (99):

$$H = J \sum_i \vec{S}_i \cdot \vec{S}_{i+1} + J_3 \sum_i \vec{S}_i \cdot \vec{S}_{i+3}. \qquad (107)$$

Fig. 3(c) shows that the apparent logarithmic growth of $D_z(t)$ disappears when $J_3 \neq 0$. (*3*) In the above models, the Hamiltonian is time independent and therefore the energy is always conserved under time evolution. We thus consider a random discrete dynamics with the

Hamiltonian [16]

$$H(t) = \sum_i J_i(t)\vec{S}_i \cdot \vec{S}_{i+1} , \tag{108}$$

where $J_i(t)$ is constant during the time interval $(nT, (n+1)T)$, and randomly choosen at time $t = nT$ and at every site $i$. This random dynamics still has SU(2) symmetry. We take $J_i$ to be uniformly distributed in $[W_1, W_2]$. Interestingly, in Fig. 3(d) we observe the same logarithmic growth at intermediate times as in the clean system.

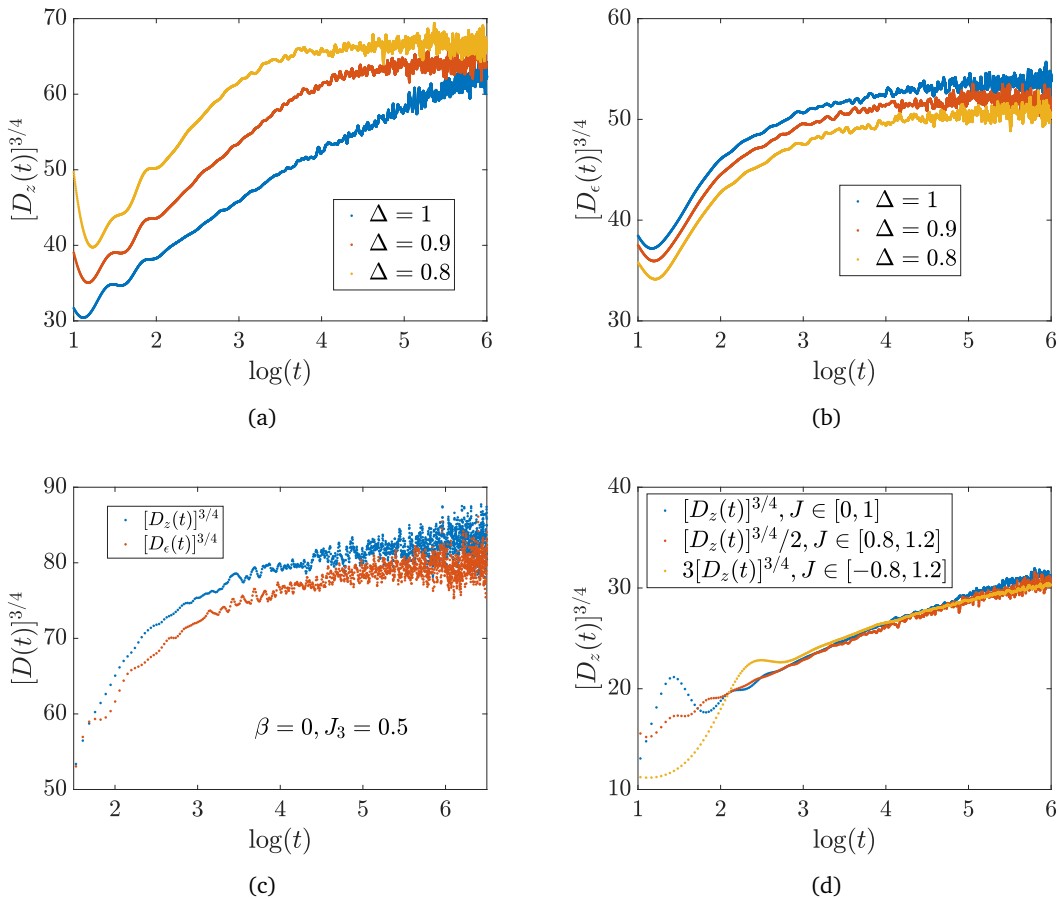

Figure 3: (a) $[D_z(t)]^{3/4}$ vs $\log t$ in XXZ model with various values of $\Delta$ at $\beta = 0$. (b) $[D_\epsilon(t)]^{3/4}$ vs $\log t$ in XXZ model with various $\Delta$ at $\beta = 0$. (c) $[D(t)]^{3/4}$ vs $\log t$ in Heisenberg model with 3rd nearest neighbor interaction and coupling coefficient $J_3 = 0.5$. (d) $[D(t)]^{3/4}$ vs $\log t$ with random dynamics with time interval $T = 0.2$. We take the same time interval in Fig. 4 and Fig. 5.

## 6.3 Symmetry sectors

We have thus seen evidence that could be interpreted as compatible with the claims of [32], although not conclusively so. However, up until now, we have only studied the canonical conservation laws of spin and energy. If the theory of [32] is correct, then there is an unambiguous numerical test: an emergent hydrodynamic mode must arise in the symmetry sector of $\tau$, defined in (102). More broadly, the analysis of Sec. 5 shows that, in order to have logarithmic enhancements of diffusion, additional conserved charges are required. If no additional

Table 1: The predicted decay rates $\gamma$ (corresponding operators denoted in parentheses) which correspond to the leading order hydrodynamic decay modes in each channel, assuming that energy is not conserved. $n^A$ denotes the conserved SU(2) charge density.

| PT | adjoint | trivial | spin-2 ($xy$) |
|---|---|---|---|
| ++ | $\frac{5}{2}$ ($\nabla^2 n^A$) | 1 ($n^A n^A$) | 1 ($n^x n^y$) |
| +− | $\frac{1}{2}$ ($n^A$) | 3 ($\nabla n^A \nabla n^A$) | 3 ($\nabla n^x \nabla n^y$) |
| −+ | $\frac{3}{2}$ ($\nabla n^A$) | 2 ($n^A \nabla n^A$) | 2 ($n^x \nabla n^y + n^y \nabla n^x$) |
| −− | $\frac{3}{2}$ ($\nabla n^A$) | 2 ($n^A \nabla n^A$) | 2 ($n^x \nabla n^y + n^y \nabla n^x$) |

charges are present, our hydrodynamic effective field theory predicts conventional spin and energy diffusion.

We have have looked at the autocorrelation function (104) for $\mathcal{O}_i$ belonging to 12 different symmetry sectors, corresponding to operators which are even or odd under parity (P) and time reversal (T), along with operators in the three "simplest" real representations of SU(2): **1** (scalar/spin 0), **3** (adjoint/spin 1), and **5** (spin 2 traceless/symmetric). The following 12 operators transform in the appropriate representations of the three groups above: labeling operators with their PT $= \pm\pm$ eigenvalues: $\mathcal{O}_{PT}$ for **1**; $\mathcal{O}^A_{PT}$ for **3**; $\mathcal{O}^{xy}_{PT}$ for **5** (for simplicity we only compute one component of the transverse traceless symmetric tensor),

$$\mathcal{O}_{++} = S_i \cdot S_{i+1}, \tag{109a}$$

$$\mathcal{O}^A_{++} = S^A_i[S_i \cdot (S_{i+1} \times S_{i+2}) + S_i \cdot (S_{i-1} \times S_{i-2})], \tag{109b}$$

$$\mathcal{O}^{xy}_{++} = (S^x_i S^y_{i+1} + S^y_i S^x_{i+1}) S_i \cdot S_{i+1}, \tag{109c}$$

$$\mathcal{O}_{+-} = S_i \cdot (S_{i+2} \times S_{i+3} + S_{i-2} \times S_{i-3}), \tag{109d}$$

$$\mathcal{O}^A_{+-} = S^A_i, \tag{109e}$$

$$\mathcal{O}^{xy}_{+-} = (S^x_{i-1} S^y_{i+1} + S^y_{i-1} S^x_{i+1}) S_i \cdot (S_{i+2} \times S_{i+3} + S_{i-2} \times S_{i-3}), \tag{109f}$$

$$\mathcal{O}_{-+} = (S_i \cdot S_{i+2})(S_{i+1} \cdot S_{i+4}) - (S_i \cdot S_{i-2})(S_{i-1} \cdot S_{i-4}), \tag{109g}$$

$$\mathcal{O}^A_{-+} = \epsilon^{ABC} S^B_i S^C_{i+1}, \tag{109h}$$

$$\mathcal{O}^{xy}_{-+} = (S^x_{i-1} S^y_{i+1} + S^y_{i-1} S^x_{i+1})[(S_i \cdot S_{i+2})(S_{i+1} \cdot S_{i+4}) - (S_i \cdot S_{i-2})(S_{i-1} \cdot S_{i-4})], \tag{109i}$$

$$\mathcal{O}_{--} = S_i \cdot (S_{i+1} \times S_{i+2}), \tag{109j}$$

$$\mathcal{O}^A_{--} = (S^A_{i+1} - S^A_{i-1})(S_{i+1} \cdot S_{i-1}), \tag{109k}$$

$$\mathcal{O}^{xy}_{--} = (S^x_i S^y_{i+2} + S^y_i S^x_{i+2}) S_i \cdot (S_{i+1} \times S_{i+2}). \tag{109l}$$

Note that $\mathcal{O}_{--}$ is a proxy for $\tau$.

### 6.3.1 No energy conservation

We begin by summarizing the hydrodynamic predictions for the decay exponents $\gamma$ defined by

$$C_\mathcal{O}(t) \sim t^{-\gamma}. \tag{110}$$

Due to nonlinear hydrodynamic fluctuations, we expect that every $\gamma$ is finite, even if $S^A$ is the only hydrodynamic degree of freedom. Indeed, we expect that without fine tuning, the decay of all of these operators will be set by the lowest possible dimension of a product of hydrodynamic operators transforming in the appropriate representation.

Predictions for the exponents $\gamma$ can be obtained from fluctuating hydrodynamics by writing constitutive relations for the operators in Eq. (109). This strategy was advocated in the context

of quantum field theories in [76] (where the symmetry used was spatial rotation $SO(d)$) and was implicitly used in [77] for a $\mathbb{Z}_2$ symmetry; it will be particularly powerful in the present context because both the internal symmetry SU(2) and discrete spatial symmetries P, T can be used. Let us illustrate one of these constitutive equations, for the operator $\mathcal{O}_{--}$ in Eq. (109j):

$$\mathcal{O}_{--} = \lambda\nabla(n^A n^A) + \lambda'\nabla^3(n^A n^A) + \lambda''\epsilon^{ABC}n^A\nabla n^B\nabla^2 n^C + \cdots, \tag{111}$$

where $\cdots$ denotes less relevant contributions, and $\lambda$, $\lambda'$, $\lambda''$ are nonuniversal coefficients that may depend on the microscopic operator $\mathcal{O}_{--}$. Given that $\nabla$ has dimension 1 and $n^A$ has dimension $1/2$ (see discussion below eq. (74)), the first term in (111) has dimension 2 and leads to the predicted decay rate $\gamma = 2$ in Table 1; the next two terms have dimension 4 and $9/2$. Note that the first two terms are even under T – we emphasize that terms with derivatives do *not* need to manifestly obey T, because derivative corrections in hydrodynamics generically break T explicitly, as in Fick's law: $J^A = -D\nabla n^A$ relates a T-odd quantity $J^A$ to a T-even quantity $n^A$ via a gradient. However, we do need to impose T for terms without gradients, because they survive at equilibrium where T is a symmetry.

The constitutive relation for the spin current (59) is another example; generic operators with the same symmetry $\mathcal{O}_{-+}^A$ will have the same terms in their constitutive relations, with different non-universal coefficients in front of each term. Table 1 lists the hydrodynamic operators and predicted values of $\gamma$ for all 12 symmetry sectors. This same approach also predicts subleading corrections to (110) [51, 78]: in the absence of a particle-hole type symmetry one expects $C_{\mathcal{O}}(t) \sim t^{-\gamma}(1 + \frac{1}{\sqrt{t}} + \cdots)$.[12] Such loop corrections to leading diffusive behavior are further discussed in Sec. 6.4.

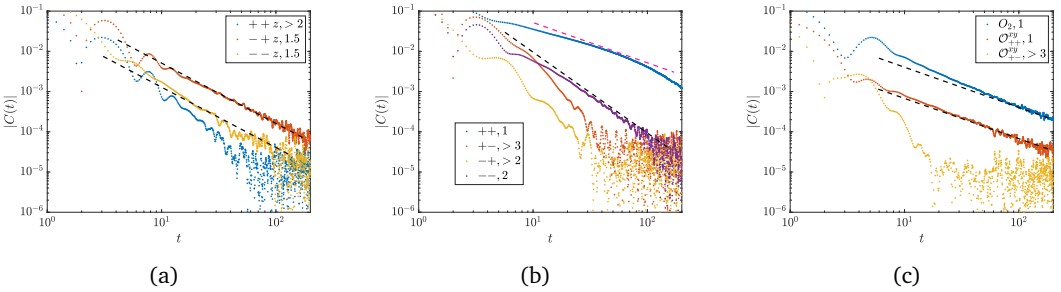

Figure 4: Correlation functions of the operators defined in Eq.(109). Here the dynamics is determined by the Heisenberg model with random coupling $J_i \in [0, 1]$ in both spatial and time directions. We only present the correlators which can show a clear power law decay in time. (a) Correlators of the adjoint representation. The black dashed line has the slope $= -1.5$. (b) Correlators of the trivial representations. The black and pink dashed lines have the slopes $-2$ and $-1$ respectively. (c) Correlators of the spin-2 representation. $\mathcal{O}_{++}^{xy}$ is the product between $O_2 = S_i^x S_{i+1}^y + S_i^y S_{i+1}^x$ and $\mathcal{O}_{++}$. Both the correlators for $O_2$ and $\mathcal{O}_{++}^{xy}$ have slope $-1$ (the same as the black dashed lines).

In Fig. 4 and Fig. 5, we present the results for $J \in [0, 1]$ and $J \in [-1, 1]$ respectively. Notice that only correlation functions with a clear power law decay are presented in the plots.

---

[12]There is a mistake in the argument of Ref. [78], which predicted a subleading correction $1/t^{1/4}$ instead of $1/t^{1/2}$. Their scaling argument shows correctly that cubic interactions of diffusive modes scale as $1/t^{1/4}$; however, two such interactions are needed in any correction to diffusion [51]. Fluctuation corrections to diffusion can be simply understood as higher powers of diffusive correlators $\langle n(t)n\rangle + \langle n(t)n\rangle^2 + \cdots \sim 1/t^{1/2} + 1/t + \cdots$. This also agrees with the scaling of corrections found in the optical conductivity [57].

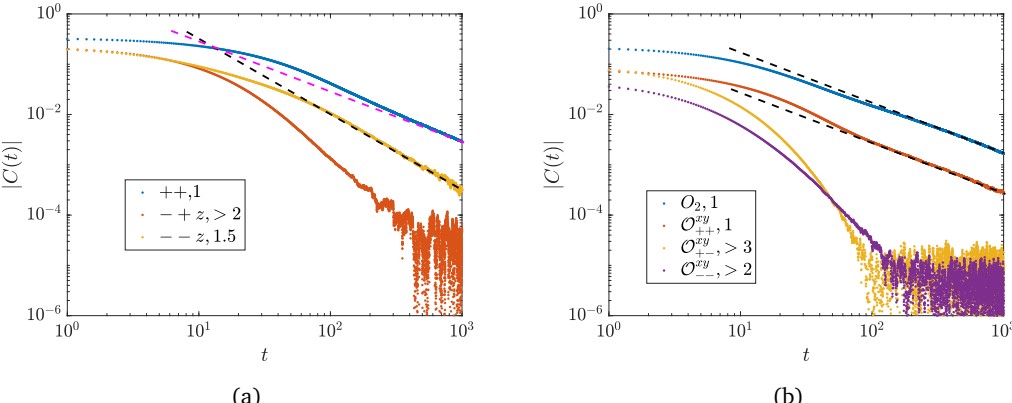

Figure 5: Correlation functions of operators defined in Eq.(109). Here the dynamics is determined by the Heisenberg model with random coupling $J_i \in [-1, 1]$ in both spatial and time directions. We only present the correlators which can show a clear power law decay in time. (a) Correlators of the adjoint and trivial representations. The black and pink dashed lines have the slope $-1.5$ and $-1$ respectively. (b) Correlators of the spin-2 representation. $\mathcal{O}_{++}^{xy}$ is the product between $O_2 = S_i^x S_{i+1}^y + S_i^y S_{i+1}^x$ and $\mathcal{O}_{++}$. Both the correlators for $O_2$ and $\mathcal{O}_{++}^{xy}$ have the slope $-1$ (the same as the black dashed lines).

The correlator for $\tau$ has power law exponent close to 2 (Fig. 4(a)), consistent with the theoretical prediction in Table 1. The rest of correlation functions with $\gamma < 2$ are also confirmed numerically. The exponent $\gamma$ is not easy to estimate once $\gamma \geq 2$, so we did not attempt a precise prediction. However, we emphasize that there is no sign of any additional emergent hydrodynamic modes, in any channel.

Some of the correlators in Fig. 5 decay faster than predicted in Table 1. This is due to an emergent $\mathbb{Z}_2$ symmetry of the model with $J \in [-1, 1]$. The equation of motion coming from the Hamiltonian (108) is invariant under spin flip $\vec{S}_i \rightarrow -\vec{S}_i$ accompanied with $J \rightarrow -J$. Additionally, for $J \in [-1, 1]$ the averaged dynamics is invariant under $J \rightarrow -J$, thus leading to invariance under spin flip. Accounting for this additional symmetry, the operators $\mathcal{O}_{-+}^A$, $\mathcal{O}_{--}^{xy}$ and $\mathcal{O}_{+-}^{xy}$ reported in Fig. 5 should be matched with $\varepsilon^{ABC} n^B \partial_x n^C$, $n^{<A} \varepsilon^{B>CD} n^C \partial_x n^D$ and $n^{<A} \varepsilon^{B>CD} n^b \partial_x^2 n^D$, where $< AB >$ denotes symmetrized traceless indices, which predicts $\gamma = 2, 2.5, 3.5$, respectively. These exponents are consistent with Fig. 5. Note that $\mathcal{O}_{--}^{xy}$ and $\mathcal{O}_{+-}^{xy}$ are matched with a cubic expression of the density $n^A$, implying that the predicted power law decays $\gamma$ are a two-loop effect of hydrodynamic fluctuations.

Note that $d_{abc} \equiv \mathrm{Tr}(T_a \{T_b, T_c\})$ vanishes for SU(2) – for bigger groups we could use $d_{abc}$ in constitutive relations to obtain slower power laws. For example, $d_{abc} n^b n^c$ is adjoint and $PT = ++$ with dimension 1 instead of 3.

### 6.3.2 Energy conservation

In Table 2 we list the hydrodynamic operators and predicted values of $\gamma$ for all 12 symmetry sectors when both energy and spin are conserved. We numerically computed these correlation functions in the Heisenberg model (99). The results are presented in Fig. 6 and Fig. 7(a) for $\beta = 0$ and $\beta = 0.5$ respectively. The power law exponents (except $\mathcal{O}_{++}^{ab}$) remain the same as we vary $\beta$. Notice that all of the decay rates are consistent with our theoretical prediction at the level of precision of the numerics. In particular, due to energy conservation, the correlator for $\tau$ operator has power law exponent close to 1.5 (See Fig. 6(a) and Fig. 7(a)). The results

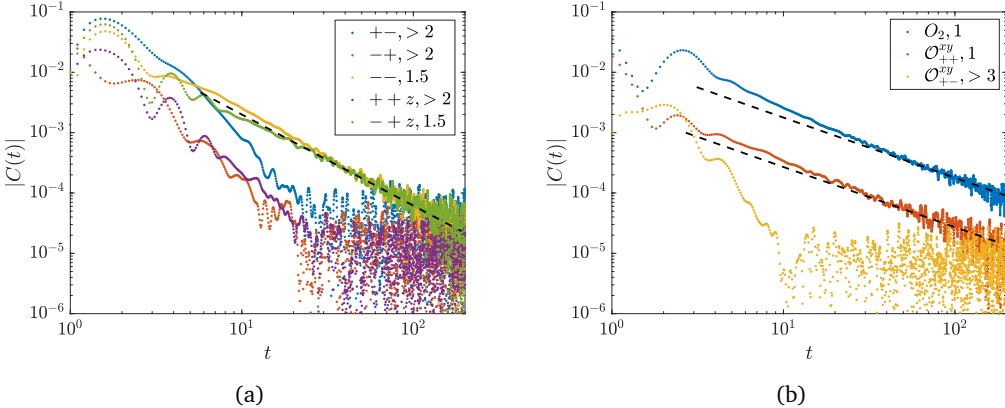

Figure 6: Correlation functions of the operators defined in Eq.(109). Here the dynamics is determined by the Heisenberg model at $\beta = 0$. We only present the correlators which can show a clear power law decay in time. (a) Correlators of the adjoint and trivial representations. The black dashed line has the slope $-1.5$. (b) Correlators of spin-2 representation. $\mathcal{O}_{++}^{xy}$ is the product between $O_2 = S_i^x S_{i+1}^y + S_i^y S_{i+1}^x$ and $\mathcal{O}_{++}$. Both the correlators for $O_2$ and $\mathcal{O}_{++}^{xy}$ have the slope $-1$ (the same as the black dashed lines).

Table 2: The predicted decay rate exponents $\gamma$ (corresponding operators denoted in parentheses) which correspond to the leading order hydrodynamic decay modes in each channel, assuming that energy is conserved. $n^A$ denotes the conserved SU(2) charge density, and $\epsilon$ denotes the conserved energy.

| PT | adjoint | trivial | spin-2 $(xy)$ |
|----|---------|---------|---------------|
| ++ | $\frac{5}{2}$ $(\nabla^2 n^A)$ | $\frac{1}{2}$ $(\epsilon)$ | 1 $(n^x n^y)$ |
| +- | $\frac{1}{2}$ $(n^A)$ | $\frac{5}{2}$ $(\nabla^2 \epsilon)$ | 3 $(\nabla n^x \nabla n^y)$ |
| -+ | $\frac{3}{2}$ $(\nabla n^A)$ | $\frac{3}{2}$ $(\nabla \epsilon)$ | 2 $(n^x \nabla n^y + n^y \nabla n^x)$ |
| -- | $\frac{3}{2}$ $(\nabla n^A)$ | $\frac{3}{2}$ $(\nabla \epsilon)$ | 2 $(n^x \nabla n^y + n^y \nabla n^x)$ |

for $\mathcal{O}_{--}^z$ $\mathcal{O}_{++}^{ab}$ and $\mathcal{O}_{-+}$ are further presented in Fig. 7(b), Fig. 7(c) and Fig. 7(d) at various values of $\beta$.

We conclude that there is no discrepancy between the numerics of the Heisenberg model and our hydrodynamic EFT, which predicts vanilla spin and energy diffusion. There appears to be no emergent hydrodynamic mode with the same symmetries as $\tau$, which is at odds with the theoretical proposal that $\tau$ is an emergent hydrodynamic mode [32]. More in general, our analysis did not detect additional emergent hydrodynamic modes, in any channel. In conclusion, we found no evidence that the integrability of the continuum Heisenberg model has any consequence on the late time dynamics of the lattice model. We emphasize however that our numerics appear consistent with the presence of an apparent logarithmically-enhanced diffusion within a finite time window (see Fig. 2(c)); in this regime our results do agree with those of [32].[13] The next Section is devoted to clarify this point.

---

[13]In the numerics of [32], a larger number of states ($5 \cdot 10^5 - 10^6$) was used in the ensemble average which should ameliorate fluctuations in the data.

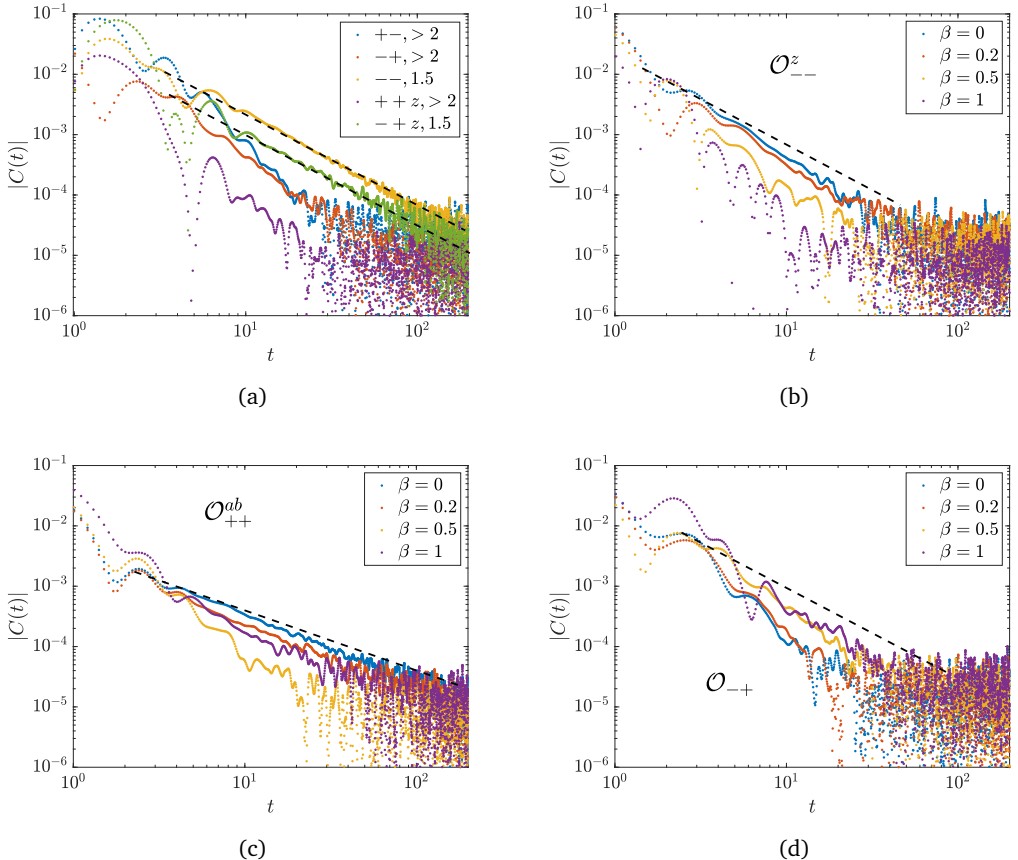

Figure 7: (a) Correlation functions for operators defined in Eq.(109). Here the dynamics is determined by the Heisenberg model at $\beta = 0.5$. We only present the correlators which can show a clear power law decay in time. The black dashed lines have the slope $-1.5$. (b) Correlation function of $\mathcal{O}^z_{--}$ at various $\beta$. The black dashed line has the slope $-1.5$. (c) Correlation function of $\mathcal{O}^{ab}_{++}$ (referred as $O_2 + +$ in Fig. 6) at various $\beta$. The black dashed line has slope $-1$. (d) Correlation function of $\mathcal{O}_{-+}$ at various $\beta$. The black dashed line has slope $-1.5$. The slope seems to approach 1.5 as we increase $\beta$.

## 6.4 Subleading corrections to hydrodynamics

We now explain the apparent logarithmic correction observed in the diffusion constant $D(t)$, within our effective field theory and numerical simulations. We start by considering a single conserved density $n$ and show qualitatively how loop corrections to the leading diffusive behavior lead to

$$C_z(t) \sim \frac{1}{\sqrt{t}} \left( 1 + \frac{a}{\sqrt{t}} \right). \tag{112}$$

A systematic study of these loop corrections using the action formalism presented in Sec. 3 can be found in Ref. [51]. The constitutive relation for a single conserved density has both nonlinear and higher derivative terms

$$J = -D\nabla n + \lambda n \nabla n + \lambda' n^2 \nabla n + \cdots + \zeta \nabla^3 n + \cdots. \tag{113}$$

The leading nonlinear term $\lambda$ is forbidden in the presence of a particle-hole symmetry $n \to -n$. Since it is a total derivative, it will not lead to corrections to transport at $k = 0$ [57]; however it

will lead to corrections to the local correlator $C_n(t, x)$, which schematically will take the form

$$C_n(t) \sim \frac{1}{t^{d/2}} \left( 1 + \frac{\lambda^2}{t^{d/2}} + \frac{\zeta}{t} + \frac{\lambda'^2}{t^d} + \cdots \right). \tag{114}$$

In $d = 1$, the leading correction to diffusion comes from loop corrections $\lambda^2/\sqrt{t}$. When $\lambda$ is forced to vanish by particle hole symmetry, the leading correction comes from higher gradient terms in the constitutive relation (113) which give $\zeta/t$; this correction cannot be forbidden by any symmetry.

For SU(2) densities, the current constitutive relation contains a nonlinear term with the same dimension as the one in (114), see Eq. (59). Written in terms of densities it takes the form

$$J^A = -D\nabla n^A + \lambda \varepsilon_{ABC} \, n^B \nabla n^C + \cdots, \tag{115}$$

and will similarly lead to $\lambda^2/\sqrt{t}$ corrections to the spin diffusion constant.

We now confirm these expectations numerically. Instead of plotting $[D_z(t)]^{3/4}$ vs $\log t$ as shown in Fig. 2(c) and Fig. 3(d), we here plot $C_z(t)\sqrt{t}$ vs $1/\sqrt{t}$ in Fig. 8 and we find that it is a straight curve when $t \gtrsim 10$, suggesting that (114) holds. Notice that in Fig. 8(a), as we increase $\beta$, the coefficient $a$ decreases and vanishes when $\beta = 1$. Furthermore, the overall magnitude of these corrections is entirely within expectations: on dimensional grounds we expect the coefficient in Eq. (112) to be $a = \alpha\sqrt{\tau_{\text{th}}}$, where[14] $\alpha \lesssim 1$. From Fig. 8(a) one finds that the correction is at most $a_{\max} \sim 1.3$ at $\beta = 0$. Estimating $\tau_{\text{th}} \sim 1$ gives $\alpha_{\max} \sim 1.3$.

Breaking the SU(2) symmetry with $\Delta \neq 1$, diffusion of spin $n = S_z$ is not expected to have the $1/\sqrt{t}$ correction because of spin-flip symmetry $n \to -n$. This is consistent with the faster equilibration of the diffusion constant in Fig. 3(a), also observed in Ref. [32]. We indeed observe in Fig. 8(c) that both the spin and energy correlators exhibit a $1/t$ correction. The $1/\sqrt{t}$ correction should reappear when studying states with a finite magnetization density $\langle S_z \rangle \neq 0$.

Finally, the result (114) can be applied to energy diffusion. At infinite temperature $\beta = 0$, we expect a particle-hole–like symmetry $\beta \to -\beta$ to forbid the leading correction [77], whereas this correction should be recovered when $\beta > 0$. Our numerical results for the energy correlator $C_\epsilon(t)$ are consistent with that prediction (see e.g. Fig. 8(c)), but unlike the results for spin they are not precise enough to sharply distinguish $1/\sqrt{t}$ and $1/t$. Hydrodynamic loop corrections to diffusion will be further studied in Ref. [80].

# 7 Conclusion

In this paper, we have derived an effective action for nonlinear fluctuating hydrodynamics in models with non-Abelian continuous symmetry groups. Our results agree with prior literature [11–13]: the non-Abelian hydrodynamics is similar to the Abelian hydrodynamics, up to the emergence of new conserved charges (one for each symmetry generator). Nonlinear fluctuations are irrelevant (in the absence of anomalies for the symmetries). These results do not depend on the thermodynamic ensemble. In particular, one should not restrict to the dynamics of a commuting (Cartan) subgroup of conserved charges, even at non-zero chemical potential. We have also discussed the fundamentally different situation of hydrodynamics with non-abelian space-time symmetries, such as rotational invariance and dipole symmetry

---

[14]Various factors can lead to the suppression of loop corrections and make $\alpha \ll 1$: large number of local degrees of freedom [79], weak dependence of transport parameters on thermodynamic potentials [52], or simply an approximate particle-hole–like symmetry (which would not suppress all loop-corrections, but only the leading ones). However we do not expect that $\alpha$ can be parametrically large.

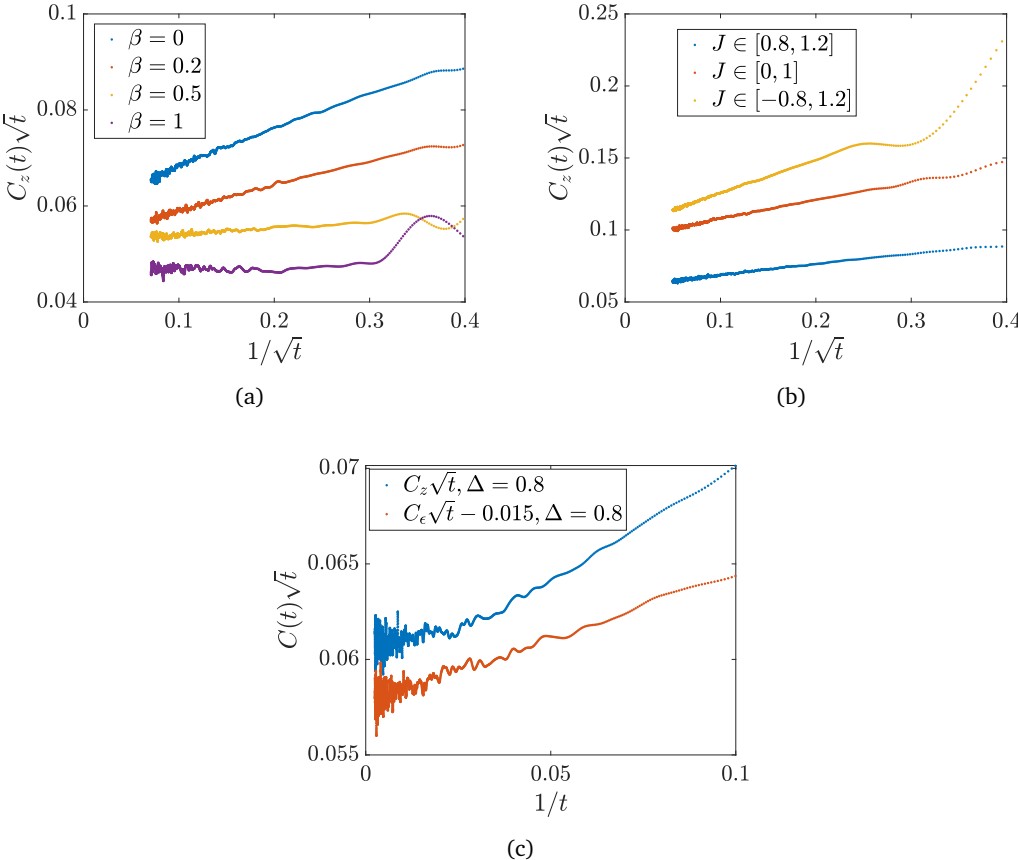

Figure 8: (a) $C_z(t)\sqrt{t}$ vs $1/\sqrt{t}$ for Heisenberg model at various $\beta$. The data for $\beta = 0.5$ and $\beta = 1$ are increased by a small constant 0.015 and 0.03 respectively. (b) $C_z(t)\sqrt{t}$ vs $1/\sqrt{t}$ with random dynamics. The data for $J \in [-0.8, 1.2]$ is subtracted by a constant 0.1. (c) $C_z(t)\sqrt{t}$ and $C_\epsilon(t)\sqrt{t}$ vs $1/t$ in XXZ model with $\Delta = 0.8$.

where, contrary to the case of internal charges, the degrees of freedom are only commuting densities. Our methods easily generalize to include the effects of conserved momentum, spontaneously broken symmetries (see [81] for related recent work), quantum anomalies, and so on. It would be interesting to include such effects in future work.

We have applied our field theoretic framework to understand the classical dynamics of SU(2)-symmetric spin chains. The field theoretic framework predicts plain vanilla diffusion for both spin and (when conserved) energy. We have tested these predictions numerically. Our numerics are compatible with conventional spin and energy diffusion, and inconsistent with proposals that fluctuating hydrodynamics leads to anomalous divergent diffusion constants. We showed that the apparent enhancement of diffusion is consistent with the presence of irrelevant hydrodynamic fluctuations associated to spin diffusion. Finally, we have proposed one finely tuned mechanism through which hydrodynamics in one dimension may contain marginally relevant operators that flow away from the diffusive fixed point. It would be interesting if such a theory can be found in any microscopic lattice model.

## Acknowledgements

We acknowledge useful conversations with Ben Doyon, Sarang Gopalakrishnan, Siddharth A. Parameswaran and Xiao-Liang Qi. RN is supported by the Air Force Office of Scientific Research under award number FA9550-20-1-0222. RN and AL are supported by the Alfred P. Sloan Foundation through Sloan Research Fellowships. LD is supported by the Swiss National Science Foundation and the Robert R. McCormick Postdoctoral Fellowship. PG is supported by a Leo Kadanoff Fellowship, by the Physical Sciences Division of the University of Chicago and by a Simons Investigators Grant.

## A  Orthogonality of macroscopic spin sectors

We construct the projectors $\mathbb{P}(n^A)$ using spin coherent states for the $L$ individual spin-$\frac{1}{2}$ degrees of freedom. For a single spin-$\frac{1}{2}$ system, we define a spin coherent state $|s\rangle$, parameterized by a unit norm vector $s^A$, by the identity

$$\langle s|\sigma^A|s\rangle = s^A. \tag{116}$$

Note that this uniquely fixes $|s\rangle$. A convenient resolution of the identity is

$$1 = \int \frac{\mathrm{d}s^A}{2\pi} |s\rangle\langle s|, \tag{117}$$

with the integral measure $\mathrm{d}s^A$ uniform on the sphere. We then pick a regulator $\delta \ll 1$ and define $\mathbb{P}(n^A)$ as follows:

$$\mathbb{P}(n^A) = \int \mathrm{d}\lambda^A \prod_{i=1}^{L} \left( \frac{\mathrm{d}s_i^A}{2\pi} \mathrm{e}^{\mathrm{i}\lambda^A s_i^A} |s_i^A\rangle\langle s_i^A| \right) \mathrm{e}^{-L(\mathrm{i}\lambda^A n^A + \delta \lambda^A \lambda^A)}. \tag{118}$$

The integral over $\lambda^A$ runs over $\mathbb{R}^3$. Intuitively, the integral over $\lambda^A$ serves to dephase all terms in the sum except for those which consist of products of spin eigenstates whose total spin expectation value is $L n^A$.

We now derive (13):

$$\mathrm{tr}\left(\mathbb{P}(n_1^A)\mathbb{P}(n_2^A)\right) = \int \mathrm{d}\lambda_1^A \mathrm{d}\lambda_2^A \prod_{i=1}^{L} \left( \frac{\mathrm{d}s_{1i}^A}{2\pi} \frac{\mathrm{d}s_{2i}^A}{2\pi} \mathrm{e}^{\mathrm{i}\lambda_1^A s_{1i}^A + \mathrm{i}\lambda_2^A s_{2i}^A} \left| \langle s_{1i}^A|s_{2i}^A\rangle \right|^2 \right) \mathrm{e}^{-L(\mathrm{i}\lambda_1^A n_1^A + \mathrm{i}\lambda_2^A n_2^A + \delta\lambda_1^A\lambda_1^A + \delta\lambda_2^A\lambda_2^A)}. \tag{119}$$

Next, we observe that

$$\left| \langle s_1^A|s_2^A\rangle \right|^2 = \frac{1 + s_1^A s_2^A}{2}, \tag{120}$$

and therefore we can carry out the $s_{1i}^A$ and $s_{2i}^A$ integrals. We start with the 2 integral, orienting the $z$-direction (in standard polar coordinates) along $\lambda_2^A$:

$$\int \frac{\mathrm{d}s_1^A \mathrm{d}s_2^A}{(2\pi)^2} e^{i\lambda_1^A s_1^A + i\lambda_2^A s_2^A} \left|\langle s_1^A | s_2^A \rangle\right|^2 = \int \frac{\mathrm{d}s_1^A}{2\pi} \int \frac{\mathrm{d}\cos\theta \, \mathrm{d}\phi}{2\pi}$$

$$e^{i\lambda_1^A s_1^A + i\lambda_2 \cos\theta} \frac{1 + s_1^x \sin\theta\cos\phi + s_1^y \sin\theta\sin\phi + s_1^z \cos\theta}{2}$$

$$= \int \frac{\mathrm{d}s_1^A}{2\pi} e^{i\lambda_1^A s_1^A} \left( \frac{\sin\lambda_2}{\lambda_2} - \frac{i}{\lambda_2}\left(\frac{\sin\lambda_2}{\lambda_2} - \cos\lambda_2\right) s_z^1 \right). \tag{121}$$

We have defined $|\lambda_1^A| = \lambda_1$, etc. Now we perform the $\lambda_1$ integral, switching the coordinates so $z$ aligns with $\lambda_1$. Similar manipulations lead to

$$\int \frac{\mathrm{d}s_1^A \mathrm{d}s_2^A}{(2\pi)^2} e^{i\lambda_1^A s_1^A + i\lambda_2^A s_2^A} \left|\langle s_1^A | s_2^A \rangle\right|^2 = 2\frac{\sin\lambda_1 \sin\lambda_2}{\lambda_1 \lambda_2} - \frac{2\lambda_1 \cdot \lambda_2}{\lambda_1^2 \lambda_2^2}\left(\frac{\sin\lambda_1}{\lambda_1} - \cos\lambda_1\right)\left(\frac{\sin\lambda_2}{\lambda_2} - \cos\lambda_2\right)$$

$$= 2\left(1 - \frac{\lambda_1^2 + \lambda_2^2}{6} - \frac{\lambda_1 \cdot \lambda_2}{9} + \right)\cdots, \tag{122}$$

where in the second line we have approximated that $\lambda_{1,2}$ are small. This approximation is justified because this integral shows up $L$ times in the product in (119), so the integral becomes evaluable by saddle point methods. Therefore, we conclude that

$$\mathrm{tr}\left(\mathbb{P}(n_1^A)\mathbb{P}(n_2^A)\right) = 2^L \int \mathrm{d}\lambda_1^A \mathrm{d}\lambda_2^A \exp\left[-L\left(\frac{\lambda_1^2 + \lambda_2^2}{2}\left(\frac{1}{3} + 2\delta\right) + \frac{\lambda_1 \cdot \lambda_2}{9} - i\lambda_1^A n_1^A - i\lambda_2^A n_2^A\right)\right]$$

$$\sim L^{-3/2} 2^L \exp\left[-L\left(\frac{9}{8}(n_1 + n_2)^2 + \frac{9}{4}(n_1 - n_2)^2\right)\right], \tag{123}$$

where in the second line we used that $\delta \ll 1$, and neglected unimportant algebraic prefactors. After a few more algebraic manipulations, we obtain (13).

## B  Green's functions

In the main text we use the retarded, symmetric and time-ordered Green's functions

$$G_{O_1 O_2}^{\mathrm{R}}(t, \vec{x}) \equiv i\theta(t)\langle[O_1(t, \vec{x}), O_2(0, 0)]\rangle \tag{124a}$$

$$G_{O_1 O_2}^{\mathrm{S}}(t, \vec{x}) \equiv \frac{1}{2}\langle\{O_1(t, \vec{x}), O_2(0, 0)\}\rangle \tag{124b}$$

$$G_{O_1 O_2}^{\mathrm{T}}(t, \vec{x}) \equiv \langle\mathcal{T}(O_1(t, \vec{x})O_2(0, 0))\rangle, \tag{124c}$$

where $O_1, O_2$ are two operators, and $\langle\cdots\rangle = \mathrm{tr}(\rho\cdots)$, where $\rho = e^{-\beta H}/\mathrm{tr}(e^{-\beta H})$ is the thermal density matrix at inverse temperature $\beta$. These are related by the identity

$$G_{O_1 O_2}^{\mathrm{T}} = G_{O_1 O_2}^{\mathrm{S}} - \frac{i}{2}(G_{O_1 O_2}^{\mathrm{R}} + G_{O_1 O_2}^{\mathrm{A}}), \tag{125}$$

and by the fluctuation-dissipation theorem

$$G_{O_1 O_2}^{\mathrm{S}}(\omega, \vec{k}) = \frac{2}{\beta\omega}\mathrm{Im}G_{O_1 O_2}^{\mathrm{R}}(\omega, \vec{k}), \tag{126}$$

where $G_{O_1 O_2}^{\mathrm{A}}(t, \vec{x}) = G_{O_2 O_1}^{R}(-t, -\vec{x})$ is the advanced Green's function.

For a conserved density $\partial_t n + \partial_i J^i = 0$ with $J^i = -D\partial_i n$, the retarded and symmetric Green's functions read

$$G_{nn}^{\text{R}}(\omega,\vec{k}) = \frac{D\chi k^2}{-\mathrm{i}\omega + Dk^2} \; , \qquad G_{nn}^{\text{S}}(\omega,\vec{k}) = \frac{2\chi Dk^2/\beta}{\omega^2 + (Dk^2)^2} \; . \tag{127}$$

The simulations of Sec. 6 probe the time-ordered Green's function $G$. Using (127) and relations (125),(126) we find

$$G_{nn}^{\text{T}}(t,\vec{x} = 0) \approx \frac{1}{(tD)^{\frac{d}{2}}} \frac{\chi}{\beta} \tag{128}$$

as $t \to \infty$, where the leading contribution comes from $G_{nn}^{\text{S}}$. This holds also for the non-Abelian densities discussed in Sec. 3, including energy conservation. The form of the diffusive pole in (127), or equivalently the scaling in (128), may change if additional charges are present, as we discuss below.

## C  Dynamical KMS symmetry

Consider a system with Hamiltonian $H$ coupled to a background source $A(t,\vec{x})$ through an operator $\mathcal{O}$. The Schwinger-Keldysh generating functional is

$$e^{\mathrm{i}W[A_1,A_2]} = \text{tr}\left[ \mathcal{T}\left( e^{-\mathrm{i}\int_{t_i}^{t_f} (H - \int_x A_1 \mathcal{O})} \right) \rho_0 \bar{\mathcal{T}}\left( e^{\mathrm{i}\int_{t_i}^{t_f} (H - \int_x A_2 \mathcal{O})} \right) \right]. \tag{129}$$

Choosing the initial state to be locally thermal $\rho_0 = e^{-\beta H_0}/\text{tr}(e^{-\beta H_0})$, and assuming that $H$ is invariant under PT as in the main text, one can derive the symmetry (50) using cylicity of the trace (130) together with the fact that $\rho_0$ generates Euclidean time translations:

$$e^{\mathrm{i}W[A_1,A_2]} = \text{tr}\left[ \mathcal{T}\left( e^{-\mathrm{i}\int_{t_i}^{t_f} (H_0 - \int_x A_1 \mathcal{O})} \right) \bar{\mathcal{T}}\left( e^{\mathrm{i}\int_{t_i}^{t_f} (H_0 - \int_x A_2 \mathcal{O}(t-\mathrm{i}\beta))} \right) \rho_0 \right] = e^{\mathrm{i}W[A_1(-t,P\vec{x}),A_2(-t-\mathrm{i}\beta,P\vec{x})]}, \tag{130}$$

where we assumed that $\mathcal{O}$ has unit eigenvalue under PT. At the level of the effective action (26), this leads to the symmetry (31a),(31b). For the current associated to a non-abelian group $G$ there is one more step. Consider a Hamiltonian $H_0$ coupled with background $A_\mu^A$ as $H = H_0 - \int d^d x A_\mu^A J^{A\mu}$. The initial state is the Gibbs ensemble

$$\rho_0 = e^{-\beta(H_0 - \mu_0^A Q^A)}, \tag{131}$$

where $Q^A$ are the generators of the algebra of $G$ and $\mu_0^A$ is the initial chemical potential. Commutation with the evolution operator is accompanied by a transformation of the current

$$\rho_0 \bar{\mathcal{T}}\left( e^{\mathrm{i}\int_{t_i}^{t_f} (H_0 - \int_x \text{tr}(A_{2\mu} J^\mu))} \right) = \bar{\mathcal{T}}\left( e^{\mathrm{i}\int_{t_i}^{t_f} (H_0 - \text{tr}(A_{2\mu} R J^\mu(t-\mathrm{i}\beta)R^{-1}))} \right) \rho_0, \tag{132}$$

where

$$R_{AB} = (e^{\beta\mu^C T^C})_{AB}, \tag{133}$$

where $T_{BC}^A$ are generators in the fundamental representation. This transformation is due to that the current operator $J^{A\mu}$ is charged under $G$. We then have the following KMS symmetry of the generating functional

$$W[A_{1\mu},A_{2\mu}] = W[A_{1\mu}(-t,P\vec{x}),(R^{-1}A_{2\mu}(-t-\mathrm{i}\beta,P\vec{x})R]. \tag{134}$$

At the level of the effective action, the hydrodynamic degree of freedom $U_2$ must also transform. Recall that the effective action must depend on the combinations (41), as demanded

by gauge invariance. For the KMS transformation to preserve such combinations $U_2$ must transform as

$$U_2(t, \vec{x}) \to U_2(-t - i\beta, P\vec{x})R,$$ (135)

leading to eq. (49).

# D   Numerical methods

In this appendix, we explain the numerical method to solve the LL equation. We consider the Heisenberg model (108) where $\vec{S}_i$ is a vector with norm $|\vec{S}_i| = 1$. The dynamics is described by the following Landau-Lifshitz equation

$$\frac{d\vec{S}_i(t)}{dt} = \vec{h}_{i,\text{eff}}(t) \times \vec{S}_i(t),$$ (136)

with the effective field

$$\vec{h}_{i,\text{eff}}(t) = J_i \vec{S}_{i+1}(t) + J_{i-1} \vec{S}_{i-1}(t).$$ (137)

Below we discuss two numerical methods which automatically conserve the magnitude of $\vec{S}_i$ under time evolution.

## D.1   Method 1

Consider a single spin governed by the Landau-Lifshitz equation,

$$\frac{d\vec{S}(t)}{dt} = \vec{h} \times \vec{S}.$$ (138)

If the field $\vec{h}$ is a constant, the solution of the above equation takes the following form:

$$\vec{S}(t) = \vec{S}_0^{\parallel} + \vec{S}_0^{\perp} \cos(|\vec{h}|t) + \frac{\vec{h}}{|\vec{h}|} \times \vec{S}_0^{\perp} \sin(|\vec{h}|t),$$ (139)

where $\vec{S}_0 \equiv \vec{S}(t = 0)$. The vector $\vec{S}$ is decomposed as

$$\vec{S} = \vec{S}^{\parallel} + \vec{S}^{\perp},$$ (140)

where $\vec{S}^{\parallel}$ is the component parallel to $\vec{h}$ and $\vec{S}^{\perp}$ is the component perpendicular to $\vec{h}$:

$$\vec{S}^{\parallel} = \left( \vec{S} \cdot \frac{\vec{h}}{|\vec{h}|} \right) \frac{\vec{h}}{|\vec{h}|}$$
$$\vec{S}^{\perp} = \vec{S} - \vec{S}^{\parallel}.$$ (141)

The many-body dynamics described by Eq. (136) can be solved by discretizing the equation into short time intervals. In each time slice $\delta t$, we first freeze the spin $\vec{S}_{2i}$ on the even sites and only evolve the spin $\vec{S}_{2i+1}$ on the odd site for a time interval $\delta t$ by using Eq. (139). The effective field for $\vec{S}_{2i+1}$ are provided by the interaction between $\vec{S}_{2i+1}$ and $\vec{S}_{2i}$. We then do the same thing for the spin on the even site for another $\delta t$. Under time evolution, both the spin and energy are conserved. This staggered timestepping method is very efficient for large scale simulation and we use it to numerically solve the Landau-Lifshitz equation in this paper. This method was first proposed in Ref. [82]. In the numerical simulation of the main text, we will set $\delta t = 0.01$, system size $L = 10^3$, and average over $10^5$ states.

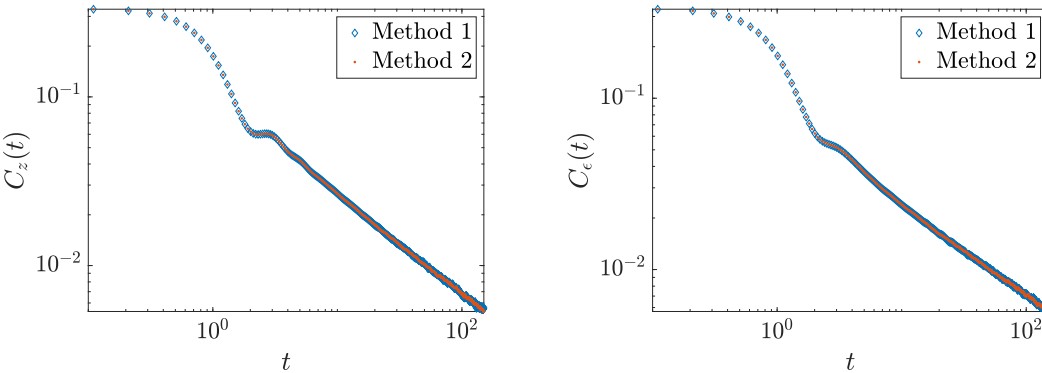

Figure 9: The $C_z$ and $C_\epsilon$ for the Heisenberg model with two different numerical methods. The system size is $L = 200$. In the first method, we take $\delta t = 0.01$.

## D.2 Method 2

We re-write Landau-Lifshitz equation in spherical coordinates:

$$\frac{\mathrm{d}\theta_i}{\mathrm{d}t} = -h_{i,x}\sin\phi_i + h_{i,y}\cos\phi_i$$
$$\sin\theta_i\frac{\mathrm{d}\phi_i}{\mathrm{d}t} = -h_{i,x}\cos\theta_i\cos\phi_i - h_{i,y}\cos\theta_i\sin\phi_i + h_{i,z}\sin\theta_i, \tag{142}$$

where $h_{i,x}$, $h_{i,y}$ and $h_{i,z}$ are the three components of the effective field caused by the interaction term in the Heisenberg model. We can solve this differential equation by the standard Runge-Kutta method. The problem in this method is that the equation is singular when $\theta = 0, \pi$. In the numerical simulation, we choose the initial state as the random state to avoid these two singular points.

## D.3 Comparison between two methods

We present our results for Heisenberg model with two different methods in Fig 9 at infinite temperature and we find that both methods give the same reliable results. In both methods, we observe the same diffusive behavior of $C_z$ and $C_\epsilon$. Furthermore, we find the same apparent

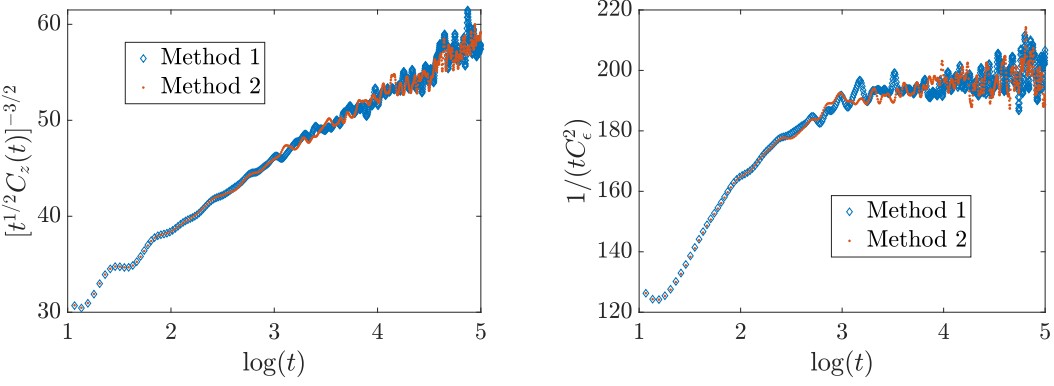

Figure 10: The scaling behavior of the diffusion constant for $C_z(t)$ and $C_\epsilon(t)$ as a function of time.

logarithmic growth of $[t^{1/2}C_z(t)]^{-3/2}$ at sufficiently early times (see Fig. 10(a)). On the other hand, as shown in Fig. 10(b), the diffusion constant for the energy correlator $C_\epsilon(t)$ saturates to a constant as time evolves.

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
