# Peer review of "Hydrodynamics in lattice models with continuous non-Abelian symmetries"

_SciPost Physics, doi:SciPost Phys. 10, 015 (2021)_

## Round 2 · Referee Report · Anonymous (Referee 1) · 2020-9-23

Report

Summary

The main technical achievement of this paper is the systematic derivation of an effective hydrodynamic theory for lattice systems with internal non-Abelian symmetry. In my view, the significance of this work is two-fold. First, it provides a much-needed summary and revision of earlier works in the same vein. Second, it is one of very few papers to bridge the gap between the effective field theory approach recently developed by the first author and collaborators, and ongoing efforts among condensed matter and statistical physicists to classify hydrodynamical behaviours in low dimensions.

Going forward, I believe this paper will play an important role in making the results of each community intelligible to those of the other and should provide a useful point of reference for future studies. However, in order for the paper to best fulfil this role, I ask that the authors clarify several points before the paper is published, especially in relation to previous literature and in Section 6. Such elisions aside, the paper is clear, mathematically concise and well-written.

Requested changes

1. p2: The points (1)-(4) emphasized by the authors seem uncontroversial to me. (1) is required by Noether’s theorem, (2) is required by equivalence of ensembles. Point (3) I found to be a succinct statement that the generic hydrodynamic behaviour is that of a “Type II” Goldstone mode with diffusive broadening, complementing the recent study of Ref. [41]. Point (4) is also to be expected, as internal symmetries differ from “mixed” symmetries in several basic respects.

I think it would be helpful to clarify, even in the introduction, why Points (1)-(4) are worth emphasizing, and how some of the more trivial confusions arose in the first place.

2. p2: I suspect that the repeated emphasis on Point (1) throughout the manuscript is based on a misconstruction by the authors of the motivation behind various works in the condensed matter literature, which study the hydrodynamics of systems with non-Abelian symmetries in physically realistic, constrained regimes, in which the number of dynamical degrees of freedom can differ from rk(G) = r. Whether or not such regimes are meaningful at asymptotically large scales certainly merits discussion, but there is not much engagement with this subtlety in the paper (beyond footnote 2).

Two concrete and well-studied examples with Abelian U(1) symmetry are the lattice Gross-Pitaevskii model (Kulkarni, Lamacraft, PRA ‘13 and Kulkarni, Huse, Spohn, PRA ‘15) and the easy-plane classical XXZ model (Ref. 62). Both of these models exhibit an emergent, low-temperature “conservation of phase difference”, which leads to robust signatures of anomalous (KPZ) broadening on accessible timescales.

For this reason, I find the remark that there are r hydrodynamic modes “…at all orders in the hydrodynamic derivative expansion…” meaningless, without further qualification. If additional slow modes need to be added by hand, then there is an error in this expansion at some finite order. I realize that there is a distinction between studying some effective theory at all orders and describing a given lattice model at all orders, but this does not seem to be acknowledged clearly in the paper (e.g. the claim of validity “to all orders” is repeated in Secs. 2.2 and 2.3.)

3. p4: The claim is repeated that
“We will see that all charges Q_A represent independent hydrodynamic degrees of freedom, and that, in a domain of length L, this conclusion will be robust to all orders in the perturbative hydrodynamic expansion in L^{−1} . This conclusion can be found in [12, 13], but appears to conflict with other literature [5]”

Could the authors clarify how, precisely, this claim conflicts with other literature? Are the earlier approaches simply incorrect, or applicable to a particular regime of times and temperatures?

As far as I understand, the claim of accuracy to “all orders” is based Eq. (2.12). While this is an appealing result, it states simply that the Hilbert space on a fluid cell may be stratified into sectors of distinct n_A, up to exponentially small corrections, i.e. is a statement about thermodynamic ensembles (see also Point 16.) The extrapolation to “hydrodynamic degrees of freedom” seems too hasty: dynamical constraints routinely emerge in low-temperature and quantum coherent settings, and such constraints need not contradict Eq. (2.12).

4. p6: Section 3 begins “In this Section we present a systematic effective field theory of hydrodynamics. One important feature of this formulation is that it captures in full generality the effects of hydrodynamic stochastic fluctuations. We will use this in later Sections to rule out any anomalous hydrodynamic transport behavior of spin diffusion for non-integrable SU(2)-invariant spin chains.”

As the authors are presumably aware, the idea of including stochastic fluctuations in nonlinear hydrodynamics dates to the 1970s, and the most thoroughly tested incarnation of the theory in this decade is the version due to Spohn and van Beijeren, in Ref. 57. I think it would be appropriate to mention these bodies of work at the start of Section 3, to allow the reader to place Refs. 32 and 33 in their proper context. Since the theory of Ref. 57 is also stochastic and effective, it would be helpful to clarify what precisely is gained by the more recent approaches. For example, is there a situation in which the approach of Ref. 57 leads to demonstrably incorrect conclusions?

I raise this point because the statement in the abstract, that the “low energy theory is a set of coupled noisy diffusion equations”, is simply what is expected on general grounds of symmetry and linear response theory. It is therefore consistent with the theory of Ref. 57 (as noted explicitly around Eq. 2.10 of Ref. 62). I would argue that anomalous transport in non-integrable, isotropic chains was already “ruled out” in Ref. 62: what remained to be explained is why it is seen at all!

5. p7: It is not clear to me what prediction the authors’ effective theory makes for lattice systems with energy conservation and internal U(1) symmetry (presumably normal diffusion for both). I would like to see a discussion of this in relation to the above mentioned works on emergent KPZ physics in the lattice GPE and classical XXZ models.

6. p9: It was not clear to me when the discussion began to specialize to SU(2). I was also curious as to how the counting in Eq. (3.29) and (3.30) generalizes, specifically whether its difficulty depends, in a simple way, on the rank and dimension of the underlying Lie algebra.

7. p9: Eq. 3.36. I think it would be helpful to describe the physics of various terms in this equation, seeing as it aspires to a universal description of hydrodynamic spin currents in lattice systems, and is therefore a significant result of the paper.

8. p10, top. It is stated that “and their coefficients can have unrelated values. This will not lead to qualitative changes in the predictions discussed below”. However, suppose lambda is the dominant term. One qualitative change, peculiar to non-Abelian symmetry, occurs in all dimensions: the hydrodynamics of spin becomes approximately norm-preserving, and the number of hydrodynamic degrees of freedom is approximately reduced by one. Further, in d=1, the hydrodynamics in this regime is near-integrable, regardless of the underlying microscopic dynamics. I think the existence of such unusual regimes within the leading-order effective theory merits comment, and is related to the subtleties arising in Section 6.

9. p12: it is stated “This same singularity was found in Ref. [42], where high-temperature non-analyticities in transport of many-body chains were first discovered. In that case energy conservation was crucial in order to have this effect. Here, the nonlinearity associated to λ is present even in systems which break energy conservation; the crucial ingredient is the presence of multiple densities”.

This is slightly inaccurate: it is already clear from the treatment of Ref. [42] that multiple densities are the “crucial ingredient” to achieve this effect at relatively low order in mode-coupling theory.

10. Section 4: I found the arguments in this section elegant, but a little confusing in relation to the rest of the paper. In 4.1, the non-commutativity of the generators was only noted at the end of the subsection, and it was not clear what symmetry group one should keep in mind. In 4.2, the lack of commutativity between rotations and translations in continuous space was alluded to (apparently the lattice of the title had been discarded). But in the continuum, it is expected that vorticity conservation is a consequence of the Euler equation.

A global aspect of the presentation I find peculiar is the overall neglect of simple dynamical constraints that generically emerge in realistic systems (e.g. phase coherence, spin coherence) and confound simple intuitions about the counting of conservation laws, at the expense of a detailed discussion of rather exotic dipole-type constraints.

11. Section 6. Let me first say that I find the analysis in this Section 6 a thorough and valuable contribution to this old debate. At the same time, the discussion of others’ recent studies of the same problem misses several important aspects of their motivation. I want to emphasize that on a qualitative level, there is little disagreement between the recent proposal that the torsional mode is responsible for the observation of anomalous transport at short times, and the argument in Sec. 6.4: both proposals ultimately attribute this anomalous behaviour to the reactive, norm-conserving piece of the spin current in Eq. 6.17.

One subtlety that I think is missing from the discussion of Sec. 6 is the fact that the continuum limit of the classical Heisenberg model is integrable. The authors correctly mention “dangerous irrelevance”, but quantifying the “danger” is precisely what previous studies were trying to achieve. It should be noted that this aspect of the debate is analogous to the decades of discussion on the Fermi-Pasta-Ulam-Tsingou chain, which similarly has an integrable continuum limit (the KdV equation), with subtle consequences for ergodicity of the FPUT dynamics.

Finally, I want to mention that the physics at issue is similar to the emergent constraints arising in the lattice Gross-Pitaevskii model mentioned above: in particular, the continuum Landau-Lifshitz equation maps exactly to the continuum NLSE, with the torsion variable mapping to the differential phase. The existence of a hydrodynamic regime for the torsional mode thus has an established precedent in the literature on non-linear fluctuating hydrodynamics.

12. p17: “Two specific possible flaws in this argument are that: (1) it appears to rely on a breakdown of ergodicity, due to the time dependence in (6.5), yet we are most interested in looking at diffusion in equilibrium correlators; (2) it has been emphasized in Ref. [62] that the continuum limit of the Heisenberg model does not apply at infinite temperature, as umklapp processes cannot be ignored.”

I think Point (1) could be better phrased. Ref [21], based on Ref [61], proposed a hydrodynamic equation that, if trusted for arbitrarily long times, implies a logarithmic divergence of D. There are multiple possibilities:
a. The dynamics of the classical Heisenberg model is not fully ergodic.
b. The description of Ref. [21] is approximately correct, up to some crossover time at which normal diffusion is restored.
c. The description of Ref. [21] is incorrect, as tau is not a hydrodynamic mode.

As far as I can tell, the numerics presented are not inconsistent with viewpoint (b) and the authors are in favour of viewpoint (c). My own view is that more work needs to be done to rule out even (a) conclusively (e.g. the model has non-trivial exact solutions), and that this will require techniques beyond hydrodynamics.
Regarding point (2), I think it should be mentioned that for the spin-1/2 XXX model, the continuum limit of the Heisenberg model does seem to apply at infinite temperature. This is not an intuitive result, but its microscopic derivation was provided recently by de Nardis et al., PRL 125, 070601. From this perspective, the authors’ Point (2) amounts to the claim that microscopic integrability suppresses Umklapp processes – why is this the case?

13. p17: “From the perspective of our effective theory, we make two general comments: (1 ) SU(2) symmetry is not sufficient to render τ a hydrodynamic mode, and even if the effects of the lattice were small, they are expected to be “dangerously irrelevant”, breaking conservation laws and qualitatively changing the character of hydrodynamics. (2 ) We can easily test for whether τ is hydrodynamic on the lattice in numerical simulations. As detailed below, we find no evidence that τ is long-lived.”

Again, I take issue with the wording here. Regarding Point (1): the proposal of tau as a hydrodynamic mode in Ref. [17] arose from considering fluctuating hydrodynamics about a frozen, classical, ferromagnetic spin-wave background. Existence of such stationary backgrounds is a much stronger condition than SU(2) symmetry, and was implicit in the treatments Refs. [17,21]. In particular, there is no expectation that the models studied in Eq. 6.10, and Figs. 4 and 5, exhibit a torsional mode: in the language of Ref. [21], the continuum limit of the Hamiltonian evolution is not even defined.

Regarding Point (2), I disagree that one can “easily test” whether tau is hydrodynamic from the numerical simulations performed; please see below.

14. p.20 “there is an unambiguous numerical test: an emergent hydrodynamic mode must arise in the symmetry sector of τ , defined in (6.4).”

I do not agree that this diagnostic is unambiguous. The derivation of tau suggests that it is a long-wavelength, highly extended degree of freedom, and I am sceptical that it could be probed by the naïve point splitting (6.11j).

For example, consider making this claim in the context of the (quantum) spin-1/2 XXZ model. There (6.11j) is actually a conserved charge density of the model (the energy current), and therefore a hydrodynamic mode. However, no-one is claiming that the energy current in the XXZ chain should have a superdiffusive component. Yet in this case, superdiffusion has been conclusively traced to the validity of a large-scale description in terms of the Landau-Lifshitz equation (de Nardis et al., PRL 125, 070601).

All this is to say that the analogous discussion for the spin-1/2 XXZ model implies that the emergence of a hydrodynamic torsional mode at long length scales need not have anything to do with the local operator in (6.11j). As a point of principle, this is unrelated to whether or not the underlying model is integrable, and contradicts the view that the proposed numerical test is "unambiguous".

15. p.21 “This is distinct from the theoretical proposal in [21], which requires an emergent hydrodynamic mode for τ operator…However, we emphasize that there is no sign of any additional emergent hydrodynamic modes, in any channel.”

The proposal of Ref. [21] is simply not applicable in this case, see Point 13.

16. p.27, eq. A7. “we have approximated that λ1,2 are small”: why is this admissible? The integration over these variables is unbounded. And what is the precise “limit” being taken in A8? I raise these points because Eq. 2.12 is advertised as a non-perturbative result to “all orders”, but the surrounding discussion is not correspondingly precise.

  • validity: -
  • significance: -
  • originality: -
  • clarity: -
  • formatting: -
  • grammar: -

Author:  Paolo Glorioso  on 2020-11-19  [id 1050]

(in reply to Report 1 on 2020-09-23)

The reply to the reports of the three referees is included in the attachment.

Attachment:

reply.pdf

---

## Round 2 · Referee Report · Anonymous (Referee 2) · 2020-10-3

Strengths

- Comprehensive study of hydrodynamics of systems with non-abelian flavour symmetries

- A fair and unifying account of the related literature: provides a good explanation of some of the recent "surprising" results

Weaknesses

- The technical (theoretical) part is hard to follow, uses a lot of jargon often at the expense of conciseness of presentation

- Notation in the technical part is often confusing

Report

This manuscript essentially contains two, almost separate papers (parts). In the first part, the authors derive a consistent hydrodynamic theory (hydrodynamic action "derived" from Schwinger-Keldysh action) for systems with non-abelian global flavor symmetries. The main conclusion here is that the non-abelian symmetry group gives you a number of conserved charges which is effectively equal to the number of generators (and not the rank) of the group. Furthermore, the paper elaborates on non-abelian systems with additional "fractonic" constraints on globally conserved dipole moments, where the hydrodynamic theory gives a trivial, frozen dynamics (or no dynamics at all!).

In the second , numerical part (section 6), the paper gives a careful numerical survey of spin (and energy) transport in non-integrable SU(2) symmetric classical spin chains, in particular in Heisenberg spin chain and its close relatives. This is particularly stimulated by a recent claim of Ref. [21] of anomalous logarithmic corrections (in time) of spin diffusion constant in non-integrable (chaotic) SU(2) symmetric systems, in particular in the classical Heisenberg spin chain. The authors find a similar qualitative effects as [21], but consistently with the previous theoretical analysis they conclude that the spin diffusion constant asymptotically saturates, although much slower than for energy diffusion or for generic non-symmetric chaotic systems, with corrections O(1/t^0.5) rather than O(1/t) (as in generic case). I find this explanation convincing, and almost settling the controversy raised by Ref. [21].

Overall, I think this work is a valuable addition to the literature and I highly recommend its publication in SciPost Physics.

Requested changes

- page 2: "focus in on" (delete "in")

- The manuscript quotes *most of* recent literature on anomalous transport in (integrable & non-integrable) classical and quantum spin chains with SU(2) symmetry: Ref. [16-30]. I do not understand the order of this quotation, it is not chronological, but also not in relevance or importance? (I would suggest chronological order though, which also suggests causal relations between references). I also believe that some important related references are missing in this list, e.g:
PRL 106 , 220601 (2011): The very first reference which observed anomalous spin transport in SU(2) quantum Heisenberg spin chain,
PRL 111 , 040602 (2013),
JSP 179, 110 (2020),
SciPost Phys. 9, 038 (2020),
PRL 125, 070601 (2020)

- page 6: I find the wording "effective field theory of hydrodynamics" a bit strange, hydrodynamics is a field theory !?

- Notation for time-and space component of charge/current vector, e.g. J^\mu=(J^t,J^i) looks quite confusing at places. t and i often resemble running time-space variables. Wouldn't (J^0,J^1) be clearer?

- Introducing the fields \varphi_{1,2} in (3.3) is not well explained. How do they relate to \lambda_{1,2} in (3.2)? This part I find really difficult to read, maybe the presentation and be more streamlined and made more concise?

- After Eq. (3.8b), \eta and \eta_\mu are introduced as "PT eigenvalues". It needs to be better explained, the eigenvalues are actually (-1)^\eta..., what values do \eta take?

- What is a "dissipative superfluid"?

- After (3.33), what does the value 0 of the second indext of EM field tensor F_{i0} mean, before this was designated as "t", see my comment above?

- After (3.34): It sounds nontrivial to me that you can annul three terms by varying 2 parameters (\mu, \beta), is this correct? (of course there might be a nontrivial relation among these terms)

- (3.46,3.47) and the text in between: I do not understand what indices 0 and 3 (which are sometimes superscript and sumetimes subscript) mean there? (has to do with flavor?)

- Last eq. of (3.48), I guess the indices "I" and "J" need to be the same there?

- (4.1), I guess x_i as a space component of space-time point x, or? But why does the integration variable then read (only) "x"? Again, notation could be made much clearer in this part!

- Subsection (4.3): The authors say that bringing dipole-moment (fractonic) constraints bring "nothing new" to the game. But they find that this implies the hydrodynamics to be "frozen", right?

So this is a rather remarkable result to me!

- In the same paragraph the authors say that G has to be "simple" and a few lines below that it needs to be "semisimple". Please clarify.

- Eq. (5.9): I do not understand the meaning of of the first equation, I guess this is a low frequency dependence of D? but then writing log(\omega)^0.5 does not make much sense (we need to put absolute values twice at least).

---

## Round 2 · Referee Report · Anonymous (Referee 3) · 2020-10-7

Strengths

1- Very thorough and comprehensive discussion of non-Abelian hydrodynamics

2- Timely topic

3- Provides a simple and believable explanation for recent surprising numerical observations.

Weaknesses

1- One of the main sections (section 3) is a bit technical.

2- This is not really a weakness, but the paper does read like independent sections which could almost belong to different papers.

3- One of the main mysteries is left open. The authors' explanation for the apparent anomalously-large diffusion constant observed in numerics relies on unusually large long time tails. While this is a plausible scenario that appears to be consistent with numerics, it's still puzzling why irrelevant corrections to diffusion are so large in those systems. (though I agree with the authors there's also no reason to expect such corrections to be small.)

Report

This paper is an excellent, comprehensive study of non-Abelian hydrodynamics. The main result of the manuscript is a study of fluctuating non-Abelian hydrodynamics using the field theory framework of Refs. [32,33]. The authors find that non-linear fluctuations are irrelevant, and do not lead to anomalous diffusion. Others systems with dipole conservations and fractonic symmetries are discussed, as well as possible log (marginal) corrections to diffusion due to additional charges.

The last section 6 is largely independent from the rest of the paper, and focuses on spin and energy transport in SU(2) invariant classical spin chains. Here the authors argue that recent observations of anomalous log corrections to diffusion can be explained by simple long time tails/irrelevant corrections to diffusion. I found this argument quite convincing, and all numerical results appear to be compatible with this simple scenario of vanilla diffusion with order O(1) irrelevant corrections.

Requested changes

I agree with many of the suggestions of the other 2 referees. One additional point: it would be useful to clarify in section II what parts could be obtained without the formalism of Refs. [32,33] . (Say using the Martin-Siggia-Rose framework, which I suspect will be more familiar to most readers.)

---

## Round 3 · Referee Report · Anonymous (Referee 1) · 2020-11-21

Report

I am satisfied with the changes made by the authors and the clarifications provided in their reply. I am happy to recommend the manuscript for publication.

---

## Round 3 · Referee Report · Anonymous (Referee 3) · 2020-11-26

Report

The authors have thoroughly addressed all points raised by the referees. I strongly recommend publication in SciPost.

---

## Editorial Decision

published